# MODEL-BASED OFFLINE REINFORCEMENT LEARNING WITH LOWER EXPECTILE Q-LEARNING

**Kwanyoung Park**  **Youngwoon Lee**
Yonsei University
`https://kwanyoungpark.github.io/LEQ/`

## ABSTRACT

Model-based offline reinforcement learning (RL) is a compelling approach that addresses the challenge of learning from limited, static data by generating imaginary trajectories using learned models. However, these approaches often struggle with inaccurate value estimation from model rollouts. In this paper, we introduce a novel model-based offline RL method, Lower Expectile Q-learning (LEQ), which provides a low-bias model-based value estimation via lower expectile regression of $\lambda$-returns. Our empirical results show that LEQ significantly outperforms previous model-based offline RL methods on long-horizon tasks, such as the D4RL AntMaze tasks, matching or surpassing the performance of model-free approaches and sequence modeling approaches. Furthermore, LEQ matches the performance of state-of-the-art model-based and model-free methods in dense-reward environments across both state-based tasks (NeoRL and D4RL) and pixel-based tasks (V-D4RL), showing that LEQ works robustly across diverse domains. Our ablation studies demonstrate that lower expectile regression, $\lambda$-returns, and critic training on offline data are all crucial for LEQ.

## 1 INTRODUCTION

One of the major challenges in offline reinforcement learning (RL) is the overestimation of values for out-of-distribution actions due to the lack of environment interactions (Levine et al., 2020; Kumar et al., 2020). Model-based offline RL addresses this issue by generating additional (imaginary) training data using a learned model, thereby augmenting the given offline data with synthetic experiences that cover out-of-distribution states and actions (Yu et al., 2020; Kidambi et al., 2020; Yu et al., 2021; Argenson and Dulac-Arnold, 2021; Sun et al., 2023). While these approaches have demonstrated strong performance in simple, short-horizon tasks, they struggle with noisy model predictions and value estimates, particularly in long-horizon tasks (Park et al., 2024). This challenge is evident in their poor performances (i.e. near zero) on the D4RL AntMaze tasks (Fu et al., 2020).

Typical model-based offline RL methods alleviate the inaccurate value estimation problem (mostly overestimation) by penalizing Q-values estimated from model rollouts with uncertainties in model predictions (Yu et al., 2020; Kidambi et al., 2020) or value predictions (Sun et al., 2023; Jeong et al., 2023). While these penalization terms prevent a policy from exploiting erroneous value estimates, the policy now does not maximize the true value, but maximizes the value penalized by heuristically estimated uncertainties, which can lead to sub-optimal behaviors. This is especially problematic in long-horizon, sparse-reward tasks, where Q-values are similar across nearby states (Park et al., 2024).

Another way to reduce bias in value estimates is using multi-step returns (Sutton, 1988; Hessel et al., 2018). CBOP (Jeong et al., 2023) constructs an explicit distribution of multi-step Q-values from thousands of model rollouts and uses this value as a target for training the Q-function. However, CBOP is computationally expensive for estimating a target value and uses multi-step returns solely for Q-learning, not for policy optimization.

To tackle these issues in model-based offline RL, we introduce a simple yet effective model-based offline RL algorithm, Lower Expectile Q-learning (LEQ). As illustrated in Figure 1, LEQ uses expectile regression with a small $\tau$ for both policy and Q-function training, providing an efficient and elegant way to achieve conservative Q-value estimates. Moreover, we propose to optimize both

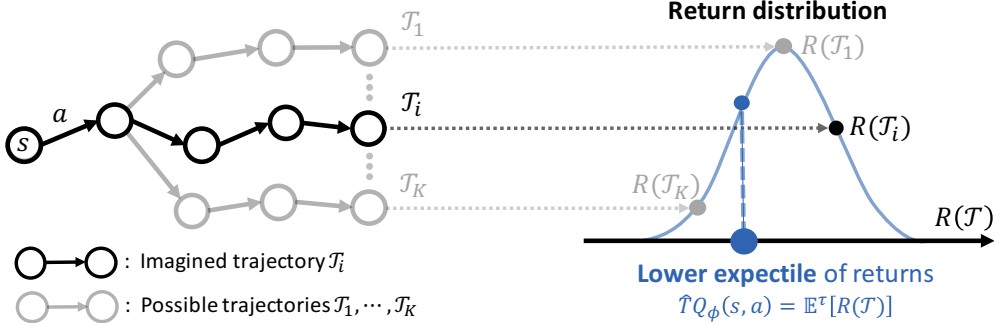

Figure 1: **Lower Expectile Q-learning (LEQ). (left)** In model-based offline RL, an agent can generate imaginary trajectories using a world model. **(right)** For conservative Q-evaluation of the policy, LEQ learns the **lower expectile** of the target $Q$-distribution from a few sampled rollouts $\mathcal{T}_i$, without estimating the entire Q-distribution with exhaustive rollouts.

policy and Q-function using $\lambda$-returns (i.e. TD($\lambda$) targets) of long (10-step) model rollouts, allowing the policy to directly learn from low-bias multi-step returns (Schulman et al., 2016).

The experiments on the D4RL AntMaze, MuJoCo Gym (Fu et al., 2020), NeoRL (Qin et al., 2022), and V-D4RL (Lu et al., 2023) benchmarks show that LEQ improves model-based offline RL across diverse domains. To the best of our knowledge, LEQ is the first model-based offline RL algorithm capable of outperforming model-free offline RL algorithms on the long-horizon AntMaze tasks (Fu et al., 2020; Jiang et al., 2023). Moreover, LEQ matches the top scores across various benchmarks, while prior methods demonstrate superior performances only for a specific domain.

## 2 RELATED WORK

**Offline RL** (Levine et al., 2020) aims to solve an RL problem only with pre-collected datasets, outperforming behavioral cloning policies (Pomerleau, 1989). While it is possible to apply off-policy RL algorithms on fixed datasets, these algorithms suffer from the overestimation of Q-values for actions unseen in the offline dataset (Fujimoto et al., 2019; Kumar et al., 2019; 2020) since the overestimated values cannot be corrected through interactions with environments as in online RL.

**Model-free offline RL** algorithms have addressed this value overestimation problem on out-of-distribution actions by (1) regularizing a policy to only output actions in the offline data (Peng et al., 2019; Kostrikov et al., 2022; Fujimoto and Gu, 2021) or (2) adopting a conservative value estimation for executing actions different from the dataset (Kumar et al., 2020; An et al., 2021). Despite their strong performances on the standard offline RL benchmarks, model-free offline RL policies tend to be constrained to the support of the data (i.e. state-action pairs in the offline dataset), which may lead to limited generalization capability.

**Model-based offline RL** approaches have tried to overcome this limitation by suggesting a better use of the limited offline data – learning a world model and generating imaginary data with the learned model that covers out-of-distribution actions. Similar to Dyna-style online model-based RL (Sutton, 1991; Hafner et al., 2019; 2021; 2023), an offline model-based RL policy can be trained on both offline data and model rollouts. But, again, learned models may be inaccurate on states and actions outside the data support, making a policy easily exploit the learned models.

Recent model-based offline RL algorithms have adopted the conservatism idea from model-free offline RL, penalizing policies incurring (1) uncertain transition dynamics (Yu et al., 2020; Kidambi et al., 2020; Yu et al., 2021) or (2) uncertain value estimation (Sun et al., 2023; Jeong et al., 2023). This conservative use of model-generated data enables model-based offline RL to outperform model-free offline RL in widely used offline RL benchmarks (Sun et al., 2023). However, uncertainty estimation is difficult and often inaccurate (Yu et al., 2021). Instead of relying on such heuristic (Yu et al., 2020; Kidambi et al., 2020; Sun et al., 2023) or expensive (Jeong et al., 2023) uncertainty

**Q-learning**          **Lower Expectile Q-learning**

$$\mathcal{L}_Q(\phi) = (Q_\phi(\mathbf{s}_t, \mathbf{a}_t) - Q_t^\lambda(\mathcal{T}))^2 \quad \Longrightarrow \quad |\tau - \mathbb{1}(Q_t^\lambda(\mathcal{T}) > Q_\phi(\mathbf{s}_t, \mathbf{a}_t))| \cdot \mathcal{L}_Q(\phi)$$

$$\nabla_\theta \mathcal{L}_\pi(\theta) = -\nabla_\theta \pi_\theta(\mathbf{s}_t) \cdot \nabla_{\mathbf{a}_t} Q_t^\lambda(\mathcal{T}) \quad \Longrightarrow \quad |\tau - \mathbb{1}(Q_t^\lambda(\mathcal{T}) > Q_\phi(\mathbf{s}_t, \mathbf{a}_t))| \cdot \nabla_\theta \mathcal{L}_\pi(\theta)$$

Figure 2: **Comparison of standard Q-learning and Lower Expectile Q-learning (LEQ).** LEQ generalizes standard Q-learning (with $\lambda$-returns $Q_t^\lambda(\mathcal{T})$) by multiplying a simple asymmetric weight "$|\tau - \mathbb{1}(Q_t^\lambda(\mathcal{T}) > Q_\phi(s_t, a_t))|$" to the Q-learning objectives. $\mathcal{T} = (\mathbf{s}_0, \mathbf{a}_0, r_0, \mathbf{s}_1, \mathbf{a}_1, r_1, \cdots, \mathbf{s}_T)$ is a model-generated trajectory and $\tau \le 0.5$ is the expectile hyperparameter controlling the degree of conservatism. When $\tau = 0.5$, LEQ reduces to standard Q-learning.

estimation, we propose to learn a conservative value function via expectile regression with a small $\tau$, which is simple, efficient, yet effective, as illustrated in Figure 2.

**Expectile regression for offline RL** has been first introduced by IQL (Kostrikov et al., 2022), which has been extended to model-based offline RL, such as IQL-TD-MPC (Chitnis et al., 2024). IQL uses *"upper expectile"* to approximate the *"max operation"* in $V(s) = \max_a Q(s, a)$ without querying out-of-distribution actions. On the other hand, our work **fundamentally differs from IQL-like approaches** in that our method uses *"lower expectile"* to get *"conservative return estimates"* from trajectories generated by potentially inaccurate model rollouts.

## 3 PRELIMINARIES

**Problem setup.** We formulate our problem as a Markov Decision Process (MDP) defined as a tuple, $\mathcal{M} = (\mathcal{S}, \mathcal{A}, r, p, \rho, \gamma)$ (Sutton and Barto, 2018). $\mathcal{S}$ and $\mathcal{A}$ denote the state and action spaces, respectively. $r : \mathcal{S} \times \mathcal{A} \to \mathbb{R}$ denotes the reward function. $p : \mathcal{S} \times \mathcal{A} \to \Delta(\mathcal{S})^1$ denotes the transition dynamics. $\rho(\mathbf{s}_0) \in \Delta(\mathcal{S})$ denotes the initial state distribution and $\gamma$ is a discounting factor. The goal of RL is to find a policy, $\pi : \mathcal{S} \to \Delta(\mathcal{A})$, that maximizes the expected return, $\mathbb{E}_{\mathcal{T} \sim p(\cdot | \pi, \mathbf{s}_0 \sim \rho)} \left[ \sum_{t=0}^{T-1} \gamma^t r(\mathbf{s}_t, \mathbf{a}_t) \right]$, where $\mathcal{T}$ is a sequence of transitions with a finite horizon $T$, $\mathcal{T} = (\mathbf{s}_0, \mathbf{a}_0, r_0, \mathbf{s}_1, \mathbf{a}_1, r_1, ..., \mathbf{s}_T)$, following $\pi(\mathbf{a}_t \mid \mathbf{s}_t)$ and $p(\mathbf{s}_{t+1} \mid \mathbf{s}_t, \mathbf{a}_t)$ starting from $\mathbf{s}_0 \sim \rho(\cdot)$.

In this paper, we consider the offline RL setup (Levine et al., 2020), where a policy $\pi$ is trained with a fixed given offline dataset, $\mathcal{D}_{\text{env}} = \{\mathcal{T}_1, \mathcal{T}_2, ..., \mathcal{T}_N\}$, without any additional online interactions.

**Model-based offline RL.** As an offline RL policy is trained from a fixed dataset, one of the major challenges in offline RL is the limited data support; thus, lack of generalization to out-of-distribution states and actions. Model-based offline RL (Kidambi et al., 2020; Yu et al., 2020; 2021; Rigter et al., 2022; Sun et al., 2023; Jeong et al., 2023) tackles this problem by augmenting the training data with imaginary training data (i.e. model rollouts) generated from the learned transition dynamics and reward model, $p_\psi(\mathbf{s}_{t+1}, r_t \mid \mathbf{s}_t, \mathbf{a}_t)$.

The typical process of model-based offline RL is as follows: (1) pretrain a model (or an ensemble of models) and an initial policy from the offline data, (2) generate short imaginary rollouts $\{\mathcal{T}\}$ using the pretrained model and add them to the training dataset $\mathcal{D}_{\text{model}} \leftarrow \mathcal{D}_{\text{model}} \cup \{\mathcal{T}\}$, (3) perform an offline RL algorithm on the augmented dataset $\mathcal{D}_{\text{model}} \cup \mathcal{D}_{\text{env}}$, and repeat (2) and (3).

**Expectile regression.** Expectile is a generalization of the expectation of a distribution $X$. While the expectation of $X$, $\mathbb{E}[X]$, can be viewed as a minimizer of the least-square objective, $\mathbb{E}_{x \sim X}[L_2(y - x)] = \mathbb{E}_{x \sim X}[\frac{1}{2}(y - x)^2]$, $\tau$-expectile of $X$, $\mathbb{E}^\tau[X]$, can be defined as a minimizer of the asymmetric least-square objective $\mathbb{E}_{x \sim X}[L_2^\tau(y - x)]$, where $L_2^\tau(\cdot)$ is defined as:

$$L_2^\tau(u) = |\tau - \mathbb{1}(u > 0)| \cdot u^2, \tag{1}$$

where $|\tau - \mathbb{1}(u > 0)|$ is an asymmetric weighting of least-squared objective with $0 \le \tau \le 1$.

We refer to a $\tau$-expectile with $\tau < 0.5$ as a *lower expectile* of $X$. When $\tau < 0.5$, the objective assigns a high weight $1 - \tau$ for smaller $x$ and a low weight $\tau$ for bigger $x$. Thus, minimizing the objective with $\tau < 0.5$ leads to a *conservative* statistical estimate compared to the expectation.

---

[1] $\Delta(\mathcal{X})$ denotes the set of probability distributions over $\mathcal{X}$.

---

**Algorithm 1** LEQ: Lower Expectile Q-learning with $\lambda$-returns

---

**Input:** Offline dataset $\mathcal{D}_{\text{env}}$, expectile $\tau \leq 0.5$, imagination length $H$, dataset expansion length $R$.

1: Initialize world models $\{p_{\psi_1}, \cdots, p_{\psi_M}\}$, policy $\pi_\theta$, and Q-function $Q_\phi$
2: Pretrain $\{p_{\psi_1}, \cdots, p_{\psi_M}\}$ on $\mathcal{D}_{\text{env}}$      $\triangleright \mathcal{L}_{\text{wm}}(\psi) = -\mathbb{E}_{(\mathbf{s},\mathbf{a},r,\mathbf{s}') \in \mathcal{D}_{\text{env}}} \log p_\psi(\mathbf{s}', r \mid \mathbf{s}, \mathbf{a})$
3: Pretrain $\pi_\theta$ and $Q_\phi$ on $\mathcal{D}_{\text{env}}$      $\triangleright$ using BC for $\pi_\theta$ and FQE (Le et al., 2019) for $Q_\phi$
4: $\mathcal{D}_{\text{model}} \leftarrow \emptyset$
5: **while** not converged **do**
6:      *// Expand dataset using model rollouts*
7:      $\mathbf{s}_0 \sim \mathcal{D}_{\text{env}}$      $\triangleright$ start dataset expansion from **any** state in $\mathcal{D}_{\text{env}}$
8:      **for** $t = 0, \ldots, R-1$ **do**
9:          $\mathcal{D}_{\text{model}} \leftarrow \mathcal{D}_{\text{model}} \cup \{\mathbf{s}_t\}$
10:         $\mathbf{a}_t = \pi_\theta(\mathbf{s}_t)$
11:         $\mathbf{s}_{t+1}, r_t \sim p_\psi(\cdot \mid \mathbf{s}_t, \mathbf{a}_t)$, where $p_\psi \sim \{p_{\psi_1}, \cdots, p_{\psi_M}\}$      $\triangleright$ sample $p_\psi$ every step
12:      *// Generate imaginary data,* $\mathcal{T} = \{(\mathbf{s}_0, \mathbf{a}_0, r_0, \cdots, \mathbf{s}_{H-1}, \mathbf{a}_{H-1}, r_{H-1}, \mathbf{s}_H)_i\}$
13:      $\mathbf{s}_0 \sim \mathcal{D}_{\text{model}}$      $\triangleright$ start imaginary rollout from **any** state in $\mathcal{D}_{\text{model}}$
14:      **for** $t = 0, \ldots, H-1$ **do**
15:         $\mathbf{a}_t = \pi_\theta(\mathbf{s}_t)$
16:         $\mathbf{s}_{t+1}, r_t \sim p_\psi(\cdot \mid \mathbf{s}_t, \mathbf{a}_t)$, where $p_\psi \sim \{p_{\psi_1}, \cdots, p_{\psi_M}\}$      $\triangleright$ sample $p_\psi$ every step
17:      *// Update critic using both **model-generated data** and **offline data***
18:      Update critic $Q_\phi$ to minimize $\mathcal{L}_Q^\lambda(\phi)$ in Eq. (7) using $\mathcal{T}$ and $\{\mathbf{s}, \mathbf{a}, r, \mathbf{s}'\} \sim \mathcal{D}_{\text{env}}$
19:      *// Update actor using only **model-generated data***
20:      Update actor $\pi_\theta$ to minimize $\hat{\mathcal{L}}_\pi^\lambda(\theta)$ in Eq. (11) using $\mathcal{T}$

---

## 4 APPROACH

The primary challenge of model-based offline RL is inherent errors in a world model and critic outside the support of offline data. Conservative value estimation can effectively handle such (falsely optimistic) errors. In this paper, we introduce Lower Expectile Q-learning (LEQ), an efficient model-based offline RL method that achieves conservative value estimation via expectile regression of Q-values with lower expectiles when learning from model-generated data (Section 4.1). Additionally, we address the noisy value estimation problem (Park et al., 2024) using $\lambda$-returns on 10-step imaginary rollouts (Section 4.2). Finally, we train a deterministic policy conservatively by maximizing the lower expectile of $\lambda$-returns (Section 4.3). The overview of LEQ is described in Algorithm 1.

### 4.1 LOWER EXPECTILE Q-LEARNING

Most offline RL algorithms primarily focus on learning a conservative value function for out-of-distribution actions. In this paper, we propose Lower Expectile Q-learning (LEQ), which learns a conservative Q-function via expectile regression with small $\tau$, avoiding unreliable uncertainty estimation and exhaustive Q-value estimation.

As illustrated in Figure 1, the target value for $Q_\phi(\mathbf{s}, \mathbf{a})$, where $\mathbf{a} \leftarrow \pi_\theta(\mathbf{s})$, can be estimated by rolling out an ensemble of world models and averaging $r + \gamma Q_\phi(\mathbf{s}', \mathbf{a}')$ over all possible $\mathbf{s}'$:

$$\hat{y}_{\text{model}} = \mathbb{E}_{\psi \sim \{\psi_1, \ldots, \psi_M\}} \mathbb{E}_{(\mathbf{s}', r) \sim p_\psi(\cdot \mid \mathbf{s}, \mathbf{a})} \left[ r + \gamma Q_\phi(\mathbf{s}', \pi_\theta(\mathbf{s}')) \right]. \tag{2}$$

This target value has three error sources: the predicted future state and reward $\mathbf{s}', r \sim p_\psi(\cdot \mid \mathbf{s}, \mathbf{a})$ and future Q-value $Q_\phi(\mathbf{s}', \pi_\theta(\mathbf{s}'))$. Thus, the target value from model-generated data, $\hat{y}_{\text{model}}$, is more prone to overestimation than the original target Q-value, $\hat{y}_{\text{env}}$, computed from $(\mathbf{s}, \mathbf{a}, r, \mathbf{s}') \sim D_{\text{env}}$:

$$\hat{y}_{\text{env}} = r + \gamma Q_\phi(\mathbf{s}', \pi_\theta(\mathbf{s}')). \tag{3}$$

To mitigate the overestimation of $\hat{y}_{\text{model}}$ from inaccurate $H$-step model rollouts, we propose to use lower expectile regression on target Q-value estimates with small $\tau$. As illustrated in Figure 2, expectile regression with small $\tau$ learns a Q-function predicting Q-values lower than the expectation, i.e., a *conservative estimate* of a target Q-value. Another advantage of using lower expectile regression is that we do not have to exhaustively evaluate Q-values to get $\tau$-expectiles as Jeong et al. (2023);

instead, we can learn a conservative Q-function with sampling:

$$L_{Q,\text{model}}(\phi) = \mathbb{E}_{\mathbf{s}_0 \in \mathcal{D}_{\text{model}}, \mathcal{T} \sim p_\psi, \pi_\theta} \left[ \frac{1}{H} \sum_{t=0}^{H} L_2^\tau (Q_\phi(\mathbf{s}_t, \pi_\theta(\mathbf{s}_t)) - \hat{y}_{\text{model}}) \right]. \tag{4}$$

Additionally, we train the Q-function with the transitions in the dataset $\mathcal{D}_{\text{env}}$, which do not have the risk of overestimation caused by the inaccurate model, using the standard Bellman update:

$$\mathcal{L}_{Q,\text{env}}(\phi) = \mathbb{E}_{(\mathbf{s},\mathbf{a},r,\mathbf{s}') \in \mathcal{D}_{\text{env}}} \left[ \frac{1}{2} (Q_\phi(\mathbf{s}, \mathbf{a}) - \hat{y}_{\text{env}})^2 \right]. \tag{5}$$

To stabilize training of the Q-function, we adopt EMA regularization (Hafner et al., 2023), which prevents drastic change of Q-values by regularizing the difference between the predictions from $Q_\phi$ and ones from its exponential moving average $Q_{\bar{\phi}}$:

$$\mathcal{L}_{Q,\text{EMA}}(\phi) = \mathbb{E}_{(\mathbf{s},\mathbf{a}) \in \mathcal{D}_{\text{env}}} \left[ (Q_\phi(\mathbf{s}, \mathbf{a}) - Q_{\bar{\phi}}(\mathbf{s}, \mathbf{a}))^2 \right]. \tag{6}$$

Finally, by combining the three aforementioned losses, we define the critic loss as follows:

$$\mathcal{L}_Q(\phi) = \beta \mathcal{L}_{Q,\text{model}}(\phi) + (1 - \beta)\mathcal{L}_{Q,\text{env}}(\phi) + \omega_{\text{EMA}}\mathcal{L}_{Q,\text{EMA}}(\phi). \tag{7}$$

## 4.2 LOWER EXPECTILE Q-LEARNING WITH $\lambda$-RETURN

To further improve LEQ, we use $\lambda$-return instead of 1-step return for Q-learning. $\lambda$-return allows a Q-function and policy to learn from low-bias multi-step returns (Schulman et al., 2016). Using $N$-step returns $G_{t:t+N}(\mathcal{T}) = \sum_{i=0}^{N-1} \gamma^i r_{t+i} + \gamma^N Q_\phi(\mathbf{s}_{t+N}, \mathbf{a}_{t+N})$, we define $\lambda$-return of an $H$-step trajectory $\mathcal{T}$ in timestep $t$, $Q_t^\lambda(\mathcal{T})$ as:[2]

$$Q_t^\lambda(\mathcal{T}) = \frac{1 - \lambda}{1 - \lambda^{H-t-1}} \sum_{i=1}^{H-t} \lambda^{i-1} G_{t:t+i}(\mathcal{T}). \tag{8}$$

Then, we can rewrite the Q-learning loss in Equation (4) with $\lambda$-return targets:

$$\mathcal{L}_{Q,\text{model}}^\lambda(\phi) = \mathbb{E}_{\mathbf{s}_0 \in \mathcal{D}_{\text{model}}, \mathcal{T} \sim p_\psi, \pi_\theta} \left[ \sum_{t=0}^{H-1} L_2^\tau (Q_\phi(\mathbf{s}_t, \pi_\theta(\mathbf{s}_t)) - Q_t^\lambda(\mathcal{T})) \right]. \tag{9}$$

## 4.3 LOWER EXPECTILE POLICY LEARNING WITH $\lambda$-RETURN

For policy optimization, we use a deterministic policy $\mathbf{a} = \pi_\theta(\mathbf{s})$ and update the policy using the deterministic policy gradients similar to DDPG (Lillicrap et al., 2016).[3] To provide more accurate learning targets for the policy, instead of maximizing the immediate Q-value, $Q_\phi(\mathbf{s}, \mathbf{a})$, we maximize the lower expectile of $\lambda$-return, analogous to our conservative critic learning in Section 4.2:

$$\mathcal{L}_\pi^\lambda(\theta) = -\mathbb{E}_{\mathbf{s}_0 \in \mathcal{D}_{\text{model}}} \left[ \sum_{t=0}^{H} \mathbb{E}_{\mathcal{T} \sim p_\psi, \pi_\theta}^\tau \left[ Q_t^\lambda(\mathcal{T}) \right] \right]. \tag{10}$$

However, because of the expectile term, $\mathbb{E}^\tau[Q_t^\lambda]$, we cannot directly compute the gradient of $\mathcal{L}_\pi^\lambda(\theta)$. To change the expectile to expectation, we use the relationship $\mathbb{E}^\tau[Q_t^\lambda] = \frac{\mathbb{E}\left[|\tau - \mathbb{1}(\mathbb{E}^\tau[Q_t^\lambda] > Q_t^\lambda)| \cdot Q_t^\lambda\right]}{\mathbb{E}\left[|\tau - \mathbb{1}(\mathbb{E}^\tau[Q_t^\lambda] > Q_t^\lambda)|\right]}$, and optimize the unnormalized version, $\mathbb{E}\left[|\tau - \mathbb{1}(\mathbb{E}^\tau[Q_t^\lambda] > Q_t^\lambda)| \cdot Q_t^\lambda\right]$. By approximating $\mathbb{E}^\tau[Q_t^\lambda]$ with the learned Q-estimator $Q_\phi(\mathbf{s}_t, \mathbf{a}_t)$, we derive a differentiable surrogate loss of Equation (10):

$$\hat{\mathcal{L}}_\pi^\lambda(\theta) = -\mathbb{E}_{\mathbf{s}_0 \in \mathcal{D}_{\text{model}}, \mathcal{T} \sim p_\psi, \pi_\theta} \left[ \sum_{t=0}^{H} |\tau - \mathbb{1}\left(Q_\phi(\mathbf{s}_t, \mathbf{a}_t) > Q_t^\lambda(\mathcal{T})\right)| \cdot Q_t^\lambda(\mathcal{T}) \right]. \tag{11}$$

Intuitively, this surrogate loss sets a higher weight $(1 - \tau)$ on a conservative $\lambda$-return estimates (i.e. $Q_\phi(\mathbf{s}_t, \mathbf{a}_t) > Q_t^\lambda(\mathcal{T})$), encouraging a policy to optimize for this conservative $\lambda$-return. On the other hand, an optimistic $\lambda$-return estimates (i.e. $Q_\phi(\mathbf{s}_t, \mathbf{a}_t) < Q_t^\lambda(\mathcal{T})$) has a less impact to the policy with a smaller weight $(\tau)$. We provide a proof in Appendix B showing that the proposed surrogate loss provides a better approximation of Equation (10) than the immediate Q-value, $Q_\phi(\mathbf{s}_t, \mathbf{a}_t)$.

---

[2]Our $\lambda$-return slightly differs from (Sutton, 1988) that puts a high weight to the last $N$-step return, $G_{t:H}(\mathcal{T})$.

[3]LEQ also works with a stochastic policy; but, a deterministic policy is sufficient for our experiments.

## 4.4 EXPANDING DATASET WITH MODEL ROLLOUTS

One of the problems of offline RL is that data distribution is limited to the offline dataset $\mathcal{D}_{\text{env}}$. To tackle this problem, we expand the dataset using simulated trajectories, which we refer to as $\mathcal{D}_{\text{model}}$ (Yu et al., 2021). To diversify the simulated trajectories in $\mathcal{D}_{\text{model}}$, we use a *noisy exploration policy*, which adds a noise $\epsilon \sim N(0, \sigma_{\text{exp}}^2)$, to the current policy and generate a trajectory of length $R$.

## 5 EXPERIMENTS

In this paper, we propose a novel model-based offline RL method with simple, efficient, yet accurate conservative value estimation. Through our experiments, we aim to answer the following questions: (1) Can **LEQ** solve diverse domains of problems, including both dense-reward tasks and long-horizon sparse-reward tasks? (2) Can **LEQ** be applied to pixel-based environments? (3) How individual components of **LEQ** affect the performance?

### 5.1 TASKS

To verify the strength of our low-bias model-based conservative value estimation in diverse domains, we test **LEQ** on four benchmarks: D4RL AntMaze, D4RL MuJoCo Gym (Fu et al., 2020), Ne-oRL (Qin et al., 2022), and V-D4RL (Lu et al., 2023). We first test on long-horizon AntMaze tasks: `umaze`, `medium`, `large` from D4RL, and `ultra` from Jiang et al. (2023), as shown in Figure 3. We also evaluate **LEQ** on locomotion tasks (Figure 4): state-based tasks from D4RL, NeoRL and pixel-based tasks from V-D4RL. Please refer to Appendix A for more experimental details.

### 5.2 COMPARED OFFLINE RL ALGORITHMS

**Model-free offline RL.** We consider behavioral cloning (**BC**) (Pomerleau, 1989); **TD3+BC** (Fujimoto and Gu, 2021), which combines BC loss to TD3; **CQL** (Kumar et al., 2020), which penalizes out-of-distribution actions; and **IQL** (Kostrikov et al., 2022), which uses upper-expectile regression to estimate the value function. For locomotion tasks, we also compare with **EDAC** (An et al., 2021), which penalizes Q-values based on its uncertainty. For pixel-based tasks, we compare with **DrQ+BC** (Lu et al., 2023), which combines BC loss to DrQ-v2 (Yarats et al., 2022); **ACRO**, which learns representations with a multi-step inverse dynamics model.

**Model-based offline RL.** We consider **MOPO** (Yu et al., 2020) and **MOBILE** (Sun et al., 2023), which penalize Q-values according to the transition uncertainty and the Bellman uncertainty of a world model, respectively; **COMBO** (Yu et al., 2021), which combines CQL with MBPO; **RAMBO** (Rigter et al., 2022), which trains an adversarial world model against the policy; and **CBOP** (Jeong et al., 2023), which utilizes multi-step returns for critic updates; **IQL-TD-MPC** (Chitnis et al., 2024), which extends TD-MPC (Hansen et al., 2022) to offline setting with IQL. For pixel-based environments, we consider **OfflineDV2** (Lu et al., 2023), which penalizes Q-values according to the dynamics errors, and **ROSMO** (Liu et al., 2023), which uses one-step model rollouts for policy improvement. LEQ follows MOBILE for most implementation details but implemented in JAX (Bradbury et al., 2018), which makes it 6 times faster than the PyTorch versions of MOBILE and CBOP. Please refer to Table 4 for detailed comparison.

**Sequence modeling for offline RL.** We consider **TT** (Janner et al., 2021), which trains a Transformer model (Vaswani, 2017) to predict offline trajectories and applies beam search to find the best

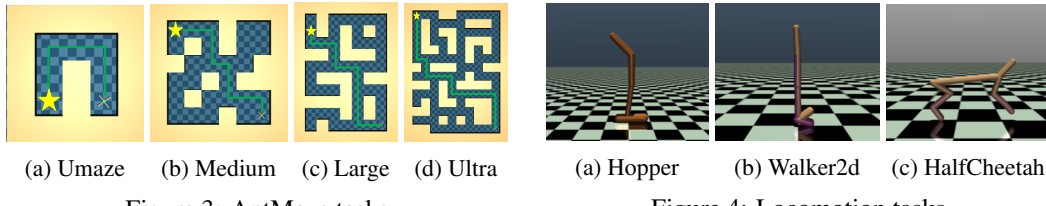

| (a) Umaze | (b) Medium | (c) Large | (d) Ultra | | (a) Hopper | (b) Walker2d | (c) HalfCheetah |

Figure 3: AntMaze tasks.          Figure 4: Locomotion tasks.

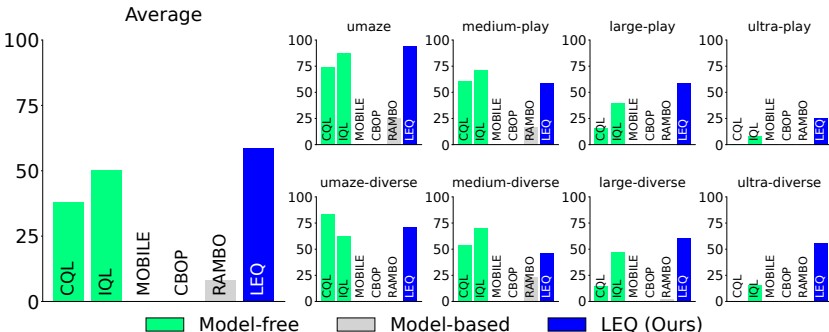

Figure 5: **AntMaze results.** Each graph displays the average success rate over 100 trials. LEQ significantly outperforms prior model-based approaches, which achieve near-zero scores, and also surpasses model-free baselines.

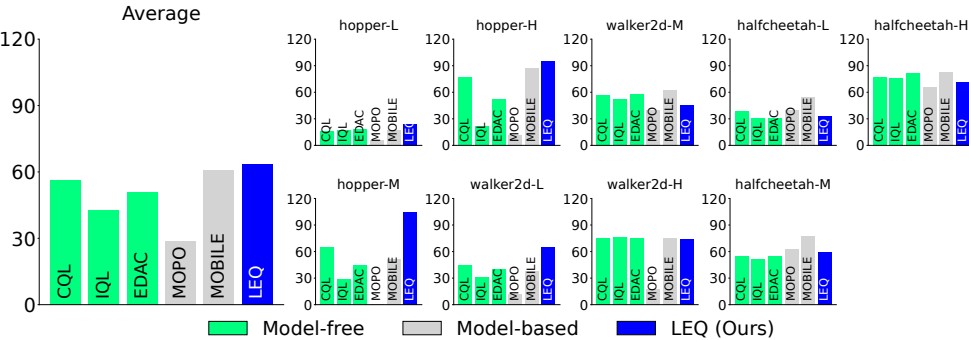

Figure 6: **NeoRL results.** Each graph displays the average normalized score over 100 trials. LEQ achieves better performance than prior methods, particularly in Hopper and Walker2d domains.

trajectory; and **TAP** (Jiang et al., 2023), which improves TT by quantizing the action space with VQ-VAE (Van Den Oord et al., 2017).

## 5.3  RESULTS ON LONG-HORIZON ANTMAZE TASKS

As shown in Figure 5, **LEQ** significantly outperforms the prior *model-based* approaches. For example, **LEQ** achieves 60.2 and 55.8 for large-diverse and ultra-diverse, while the second best method, **IQL-TD-MPC** (Chitnis et al., 2024), scores only 4.0 and 3.6, respectively. We believe these performance gains come from our conservative value estimation, which works more stable than the uncertainty-based penalization of prior works. Moreover, **LEQ** even significantly outperforms the *model-free* approaches in umaze, large, and ultra mazes, and outperforms *sequence modeling* methods, **TT** and **TAP**, which serve as strong baselines for AntMaze tasks, in the most challenging ultra mazes, showing the advantage of utilizing low-bias multi-step return on long-horizon tasks.

Despite its superior performance, **LEQ** shows high variance in the performance on antmaze-medium. We found that the medium mazes include states separated by walls that are very close to each other (denoted as red circles in Figure 7), such that all of the learned world models falsely believe the agent can pass through the walls. This incorrect prediction makes the agent to plan faster, but impossible trajectories as shown in Figure 7. We believe that this could be addressed by employing improved world models or increasing the number of ensembles for the world models.

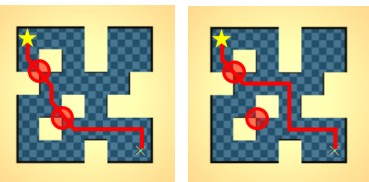

Figure 7: **Failure in medium mazes.** The agent plans impossible trajectories on certain states (red circles).

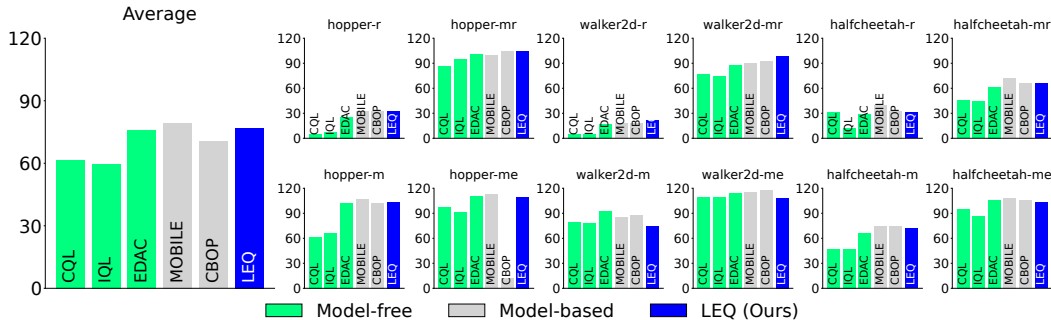

Figure 8: **D4RL results.** Each graph displays the average normalized score over 100 trials. LEQ achieves comparable results with state-of-the-art model-based and model-free offline RL methods.

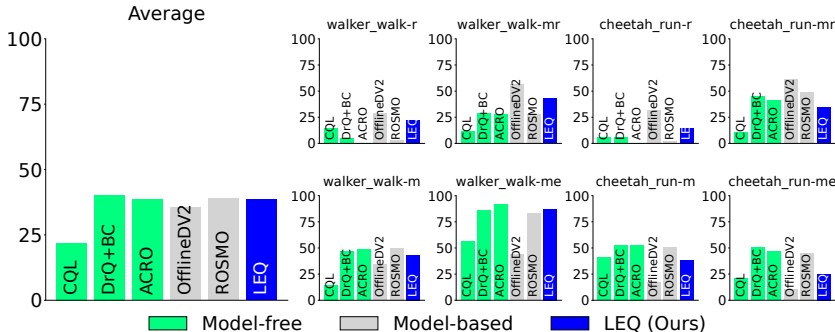

Figure 9: **V-D4RL results.** Each graph displays the average normalized score over 100 trials. LEQ shows comparable results with state-of-the-art offline visual control methods.

## 5.4 RESULTS ON MUJOCO GYM LOCOMOTION TASKS

For the NeoRL benchmark in Figure 6, **LEQ** outperforms most of the prior works, especially in the Hopper and Walker2d domains. Furthermore, for D4RL MuJoCo Gym tasks in Figure 8, **LEQ** achieves comparable results with the best score of prior works in 7 out of 12 tasks, These results show that **LEQ** serves as a general offline RL algorithm, widely applicable to various domains.

Similar to `antmaze-medium`, **LEQ** experiences high variance in MuJoCo tasks. During training, **LEQ** often achieves high performance, but then, suddenly falls back to 0, as shown in Figure 10. This is mainly because the learned models sometimes fail to capture failures (e.g. hopper and walker falling off) and predict an optimistic future (e.g. hopper and walker walking forward).

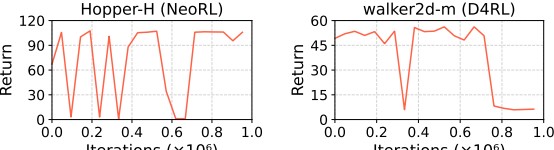

Figure 10: **High variance during training.** Our algorithm experiences oscillation due to optimistic imaginations near the initial states.

## 5.5 RESULTS WITH VISUAL INPUTS

As shown in Figure 9, **LEQ** combined with DreamerV3 (Hafner et al., 2023) performs on par with the state-of-the-art methods on V-D4RL datasets, demonstrating its scalability to visual observations. Notably, **LEQ** achieves the highest score on the `walker_walk-medium_expert` dataset among model-based methods, where **OfflineDV2** and **MOPO** struggles. **LEQ** also outperforms model-free approaches on `random` datasets and `walker_walk-medium_replay` dataset, highlighting the strength of model-based methods in datasets with diverse state-action distributions.

Table 1: **Impact of lower expectile Q-learning and $\lambda$-returns on AntMaze.** We ablate the effects of lower expectile and $\lambda$-returns on the critic and policy updates in **LEQ**. The design choices from **LEQ** are colored in blue and other options are colored in red. The results are averaged over 5 seeds.

| Design choices | | | umaze | | medium | | large | | ultra | | Total |
|---|---|---|---|---|---|---|---|---|---|---|---|
| conservatism | critic update | policy update | umaze | diverse | play | diverse | play | diverse | play | diverse | |
| Lower expectile | $\lambda$-returns | $\lambda$-returns | $94.4_{\pm6.3}$ | $71.0_{\pm12.3}$ | $50.2_{\pm39.9}$ | $46.2_{\pm23.2}$ | $58.6_{\pm9.1}$ | $60.2_{\pm18.3}$ | $25.8_{\pm18.2}$ | $55.8_{\pm18.3}$ | $461.8$ |
| Lower expectile | $H$-step | $H$-step | $93.0_{\pm3.4}$ | $60.7_{\pm10.4}$ | $46.3_{\pm32.4}$ | $0.0_{\pm0.0}$ | $57.0_{\pm25.6}$ | $33.3_{\pm43.0}$ | $0.0_{\pm0.0}$ | $0.0_{\pm0.0}$ | $290.3$ |
| Lower expectile | 1-step | $Q(\mathbf{s},\mathbf{a})$ | $89.6_{\pm4.8}$ | $37.0_{\pm32.8}$ | $55.8_{\pm28.7}$ | $29.8_{\pm24.5}$ | $34.2_{\pm13.4}$ | $49.3_{\pm9.0}$ | $42.2_{\pm13.2}$ | $35.6_{\pm13.0}$ | $373.5$ |
| Lower expectile | $\lambda$-returns | $Q(\mathbf{s},\mathbf{a})$ | $81.0_{\pm10.5}$ | $46.2_{\pm16.8}$ | $61.8_{\pm12.4}$ | $40.6_{\pm11.4}$ | $39.2_{\pm12.5}$ | $40.5_{\pm11.7}$ | $42.8_{\pm21.8}$ | $47.5_{\pm5.9}$ | $410.3$ |
| Lower expectile | $\lambda$-returns | AWR | $69.2_{\pm7.5}$ | $44.4_{\pm18.4}$ | $0.6_{\pm0.6}$ | $0.0_{\pm0.0}$ | $0.0_{\pm0.0}$ | $0.0_{\pm0.0}$ | $0.0_{\pm0.0}$ | $0.0_{\pm0.0}$ | $114.2$ |
| MOBILE | $\lambda$-returns | $\lambda$-returns | $84.3_{\pm3.5}$ | $40.3_{\pm20.4}$ | $51.3_{\pm9.0}$ | $39.7_{\pm12.5}$ | $28.3_{\pm21.5}$ | $33.7_{\pm10.0}$ | $38.0_{\pm27.1}$ | $23.3_{\pm4.9}$ | $338.9$ |
| MOBILE | 1-step | $Q(\mathbf{s},\mathbf{a})$ | $59.5_{\pm3.5}$ | $46.5_{\pm1.5}$ | $57.0_{\pm11.0}$ | $54.0_{\pm9.0}$ | $23.5_{\pm19.5}$ | $38.5_{\pm1.5}$ | $39.5_{\pm11.5}$ | $20.5_{\pm20.5}$ | $339.0$ |

## 5.6 ABLATION STUDIES

To deeply understand how **LEQ (LEQ-$\lambda$)** work, we conduct ablation studies in AntMaze environments and answer to the following five questions: (1) Is **lower expectile Q-learning** better than prior uncertainty-based penalization methods? (2) Does **lower expectile policy learning** better than existing policy learning methods? (3) Does $\lambda$-**return** help? (4) Which factor enables **LEQ** to work in AntMaze? and (5) How do **imagination length** $H$ and **data expansion** affect the performance?

**(1) Lower expectile Q-learning.** We compare our *lower expectile Q-learning* with the conservative value estimator in **MOBILE** (Sun et al., 2023), which penalizes Q-values based on the standard deviation of Q-ensemble networks. In Table 1, replacing lower expectile Q-learning with **MOBILE** decreases the success rate, both with $\lambda$-returns (461.8 vs 338.9) and without them (373.5 vs 339.0).

**(2) Lower expectile policy learning.** We also compare our *lower expectile policy learning* with AWR (Peng et al., 2019) and directly maximizing $Q(s, a)$. As shown in Table 1, **LEQ** shows better performance compared to AWR (461.8 vs 114.2) and maximizing the $Q$-values (461.8 vs 410.3).

**(3) $\lambda$-returns.** The first three rows of Table 1 show the effect of $\lambda$-*returns* in **LEQ**. Substituting $\lambda$-returns with $H$-step returns (461.8 vs 290.3) or 1-step returns (461.8 vs 373.5) significantly decreases the performance. Moreover, while **LEQ** shows better performance than **MOBILE** without $\lambda$-returns (373.5 vs 339.0), the performance of **LEQ** gets significantly better with $\lambda$-returns, compared to **MOBILE** with $\lambda$-returns (461.8 vs 338.9).

**(4) What makes model-based offline RL work in AntMaze?** **LEQ** shows outstanding performance compared to previous offline model-based RL methods, especially in large and ultra mazes. To understand which aspects of **LEQ** enabled this success, we applied its changes to **MOBILE** and analyzed the impact. The results are detailed in Table 2.

We first re-implement **MOBILE** with some technical tricks used in **LEQ** (denoted as **MOBILE**$^*$): LayerNorm (Ba et al., 2016), SymLog (Hafner et al., 2023), single Q-network, and no target Q-value clipping. However, **MOBILE**$^*$ still achieves a barely non-zero score, 14.0.

Table 2: **Impact of hyperparameters in MOBILE$^*$ on AntMaze. MOBILE**$^*$ uses the hyperparameters from **MOBILE**: $\beta = 0.95$, $\gamma = 0.99$, and $R = 5$, whereas **LEQ** uses $\beta = 0.25$, $\gamma = 0.997$, and $R = 10$. The results show that $\beta$ is the most critical hyperparameter that makes **MOBILE**$^*$ work in AntMaze.

| Hyperparams. | | | umaze | | medium | | large | | ultra | | Total |
|---|---|---|---|---|---|---|---|---|---|---|---|
| $\beta$ | $\gamma$ | $R$ | umaze | diverse | play | diverse | play | diverse | play | diverse | |
| 0.25 | 0.997 | 10 | $53.8_{\pm26.8}$ | $22.5_{\pm22.2}$ | $54.0_{\pm5.8}$ | $49.5_{\pm6.2}$ | $28.3_{\pm6.0}$ | $28.0_{\pm11.4}$ | $25.5_{\pm6.9}$ | $23.8_{\pm15.8}$ | $285.3$ |
| 0.25 | 0.997 | 5 | $74.0_{\pm6.9}$ | $3.7_{\pm2.6}$ | $54.7_{\pm27.9}$ | $28.0_{\pm9.6}$ | $18.7_{\pm18.6}$ | $8.0_{\pm9.3}$ | $9.7_{\pm8.2}$ | $9.0_{\pm3.7}$ | $205.7$ |
| 0.25 | 0.99 | 10 | $39.7_{\pm23.4}$ | $5.0_{\pm7.1}$ | $39.3_{\pm27.9}$ | $38.0_{\pm15.0}$ | $0.0_{\pm0.0}$ | $3.7_{\pm5.2}$ | $0.0_{\pm0.0}$ | $0.0_{\pm0.0}$ | $125.7$ |
| 0.25 | 0.99 | 5 | $77.0_{\pm6.4}$ | $20.4_{\pm15.7}$ | $64.6_{\pm11.1}$ | $31.6_{\pm16.9}$ | $2.6_{\pm2.8}$ | $7.2_{\pm8.9}$ | $4.6_{\pm3.0}$ | $5.0_{\pm4.6}$ | $213.0$ |
| 0.95 | 0.997 | 10 | $0.0_{\pm0.0}$ | $0.0_{\pm0.0}$ | $1.8_{\pm3.0}$ | $0.5_{\pm0.9}$ | $0.2_{\pm0.4}$ | $2.2_{\pm2.3}$ | $1.0_{\pm1.7}$ | $0.0_{\pm0.0}$ | $5.7$ |
| 0.95 | 0.997 | 5 | $0.0_{\pm0.0}$ | $0.0_{\pm0.0}$ | $7.2_{\pm4.1}$ | $1.6_{\pm2.1}$ | $9.6_{\pm7.1}$ | $5.4_{\pm4.9}$ | $0.0_{\pm0.0}$ | $1.8_{\pm2.7}$ | $25.6$ |
| 0.95 | 0.99 | 10 | $0.0_{\pm0.0}$ | $0.0_{\pm0.0}$ | $5.0_{\pm5.1}$ | $0.6_{\pm1.2}$ | $7.4_{\pm14.8}$ | $1.6_{\pm3.2}$ | $0.0_{\pm0.0}$ | $0.0_{\pm0.0}$ | $14.6$ |
| 0.95 | 0.99 | 5 | $1.0_{\pm2.0}$ | $0.0_{\pm0.0}$ | $6.4_{\pm5.5}$ | $5.0_{\pm5.0}$ | $0.8_{\pm1.6}$ | $0.8_{\pm1.2}$ | $0.0_{\pm0.0}$ | $0.0_{\pm0.0}$ | $14.0$ |

Table 3: **LEQ with different imagination length $H$ and data expansion.** A longer $H$ can mitigate critic biases, while increasing model errors, which leads to poor performance. Each number is averaged over 5 random seeds.

| Dataset | $H = 10$ (ours) | $H = 5$ | $H = 15$ | w/o dataset expansion |
|---|---|---|---|---|
| antmaze-umaze | **94.4** $_{\pm 6.3}$ | **95.2** $_{\pm 1.7}$ | **98.6** $_{\pm 0.5}$ | **97.4** $_{\pm 1.4}$ |
| antmaze-umaze-diverse | **71.0** $_{\pm 12.3}$ | 67.2 $_{\pm 9.1}$ | **70.7** $_{\pm 15.2}$ | 63.0 $_{\pm 23.2}$ |
| antmaze-medium-play | 58.8 $_{\pm 33.0}$ | 46.4 $_{\pm 31.9}$ | **76.3** $_{\pm 17.2}$ | 58.2 $_{\pm 28.0}$ |
| antmaze-medium-diverse | **46.2** $_{\pm 23.2}$ | 18.6 $_{\pm 28.7}$ | 30.3 $_{\pm 40.1}$ | 28.6 $_{\pm 33.7}$ |
| antmaze-large-play | 58.6 $_{\pm 9.1}$ | 48.6 $_{\pm 15.4}$ | **62.0** $_{\pm 9.9}$ | 56.0 $_{\pm 9.8}$ |
| antmaze-large-diverse | **60.2** $_{\pm 18.3}$ | 35.2 $_{\pm 8.7}$ | 33.0 $_{\pm 3.2}$ | **57.0** $_{\pm 4.5}$ |
| antmaze-ultra-play | 25.8 $_{\pm 18.2}$ | **54.2** $_{\pm 10.8}$ | 0.0 $_{\pm 0.0}$ | 39.2 $_{\pm 15.1}$ |
| antmaze-ultra-diverse | **55.8** $_{\pm 18.3}$ | 39.4 $_{\pm 6.1}$ | 0.0 $_{\pm 0.0}$ | 36.0 $_{\pm 12.0}$ |
| **Total** | **470.4** | 404.8 | 371.0 | 435.4 |

We found that reducing the ratio $\beta$ between the losses from imaginary rollouts and dataset transitions is key to make **MOBILE**$^*$ work (i.e. achieving meaningful performances in `umaze` and `medium` mazes, with a total score of 213.0). This adjustment also allows for a higher discount rate and longer imagination horizon, yielding the best results for **MOBILE**$^*$. We suggest that utilizing the true transition from the dataset is important in long-horizon tasks, which has been undervalued in prior works. We provide additional extensive ablation results on **LEQ** in Appendix E.

**(5) Imagination length $H$ and dataset expansion.** As shown in Table 3, the performance increases when it goes to $H = 10$ from $H = 5$, but it drops when $H = 15$. This result shows the trade-off of using the world model – the further the agent imagines, the more the agent becomes robust to the error of the critic, but the more it becomes prone to the error from the model prediction.

We also evaluate **LEQ** without the dataset expansion. In AntMaze, the results with and without the dataset expansion are similar, as shown in Table 3. On the other hand, the dataset expansion makes the policy more stable and better in the D4RL MuJoCo tasks (in Appendix, Table 20).

## 6 Limitations

Following prior work on model-based offline RL (Sun et al., 2023; Jeong et al., 2023), we assume access to the ground-truth *termination function* of a task, different from online model-based RL approaches, which learn a termination function from interactions. As shown in Table 22, Using a learned termination function instead of the ground-truth termination function results in a significant performance drop ($461.8 \rightarrow 232.6$), particularly in diverse datasets ($233.2 \rightarrow 66.8$), where termination signals are limited because the dataset is collected by navigating to randomly selected goals. While relying on the ground-truth termination function simplifies the problem, it limits the applicability of the method to scenarios where this information is readily available. Extending the proposed approach to learn terminal signals from the dataset would be an immediate next step.

## 7 Conclusion

In this paper, we propose a novel offline model-based reinforcement learning method, LEQ, which uses *expectile regression* to get a *conservative evaluation* of a policy from model-generated trajectories. Expectile regression eases the pain of constructing the whole distribution of Q-targets and allows for learning a conservative Q-function via sampling. Combined with $\lambda$-returns in both critic and policy updates for the imaginary rollouts, the policy can receive learning signals that are more robust to both model errors and critic errors. We empirically show that LEQ robustly improves the performance of model-based approaches in various domains, including state-based locomotion, long-horizon navigation, and visual control tasks.

### Acknowledgments

We would like to thank Junik Bae for helpful discussion. This work was supported in part by the Institute of Information & Communications Technology Planning & Evaluation (IITP) grant

(RS-2020-II201361, Artificial Intelligence Graduate School Program (Yonsei University)) and the National Research Foundation of Korea (NRF) grant (RS-2024-00333634) funded by the Korean Government (MSIT). Kwanyoung Park was supported by the Electronics and Telecommunications Research Institute (ETRI) grant (24ZR1100) and IITP grant (RS-2024-00509279, Global AI Frontier Lab) funded by the Korea government (MSIT). This research was supported by Korea Basic Science Institute(National research Facilities and Equipment Center) grant funded by the ministry of Science and ICT(No. RS-2024-00403860)

## ETHICS STATEMENT

Our method aims to increase the ability of autonomous agents, such as robots and self-driving cars, to learn from static, offline data without interacting with the world. This enables autonomous agents to utilize data with diverse qualities (not necessarily from experts). We believe that this paper does not have any immediate negative ethical concerns.

## REPRODUCIBILITY STATEMENT

To ensure the reproducibility of our work, we provide the full code of LEQ in the supplementary materials, along with instructions to replicate the experiments presented in the paper. We provide the experimental details in Appendix A and the proof on the derivation of our surrogate policy objective in Appendix B.

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

# A    TRAINING DETAILS

**Computing resources.**    All experiments are done on a single RTX 4090 GPU and 8 AMD EPYC 9354 CPU cores, supported by Advanced Database System Infrastructure(NFEC-2024-11-300458). For state-based environments, we use 5 different random seeds for each experiment and report the mean and standard deviation, while each offline RL experiment takes 2 hours for ours, 12 hours for MOBILE, and 24 hours for CBOP. For pixel-based environments, we use 3 different random seeds and each experiment takes 2 hours for ours and 8 hours for ROSMO.

**Environment details.**    For state-based locomotion tasks, we use the datasets provided by D4RL (Fu et al., 2020) and NeoRL (Qin et al., 2022). Following IQL (Kostrikov et al., 2022), we normalize rewards using the maximum and minimum returns of all trajectories. We use the true termination functions of the environments, implemented in MOBILE (Sun et al., 2023). For pixel-based environments, we do not normalize the rewards.

For AntMaze tasks, we use the datasets provided by D4RL (Fu et al., 2020). Following IQL (Kostrikov et al., 2022), we subtract $1$ from the rewards in the datasets so that the agent receives $-1$ for each step and $0$ on termination. We use the true termination functions of the environments. The termination functions of the AntMaze tasks are not deterministic because a goal of a maze is randomized every time the environment is reset. Nevertheless, we follow the implementation of CBOP (Jeong et al., 2023), where the termination region is set to a circle around the mean of the goal distribution with the radius $0.5$.

**Implementation details of compared methods.**    For all compared methods, we use the results from their corresponding papers when available. For IQL (Kostrikov et al., 2022), we run the official implementation with 5 seeds to reproduce the results for the random datasets in D4RL and NeoRL. For the AntMaze tasks, we run the official implementation of MOBILE and CBOP with 5 random seeds. Please note that the original MOBILE implementation does not use the true termination function, so we replace it with our termination function. For MOPO, COMBO, and RAMBO, we use the results reported in RAMBO (Rigter et al., 2022). For DMControl tasks, we replace the categorical distribution of the policy with gaussian distribution of the official ROSMO codebase, and run the experiments with sampling hyperparameter $N = 4$.

**World models.**    For state-based environments, we use the architecture and training script from OfflineRL-Kit (Sun, 2023), matching the implementation of MOBILE (Sun et al., 2023). Each world model is implemented as a 4-layer MLPs with the hidden layer size of $200$. We construct an ensemble of world models by selecting 5 out of 7 models with the best validation scores. We pretrain the ensemble of world models for each of 5 random seeds (i.e. training in total 35 world models and using 25 models), which takes approximately 5 hours in average. For pixel-based environments, we use the 12M model of DreamerV3 (Hafner et al., 2023) and pretrain the world model using its loss function. We follow the implementation of OfflineDV2 (Lu et al., 2023), training 7 ensemble of world models for stochastic latent prediction, which takes 4 hours in average. We select one model for each step, following the design choice of MOBILE, as we found that randomly choosing a model every step can make imaginary rollouts more robust to model biases, leading to better performance.

**Policy and critic networks.**    For state-based environments, we use 3-layer MLPs with size of $256$ both for the policy network and the critic network. We use layer normalization (Ba et al., 2016) to prevent catastrophic over/underestimation (Ball et al., 2023), and squash the state inputs using symlog to keep training stable from outliers in long-horizon model rollouts (Hafner et al., 2023). For pixel-based environments, we use the architecture of the 12M model in DreamerV3 models.

**Pretraining policy and critic networks.**    For some environments, we found that a randomly initialized policy can lead to abnormal rewards or transition prediction from the world models in the early stage, leading to unstable training (Jelley et al., 2024). Following CBOP (Jeong et al., 2023), we pretrain a policy $\pi_\theta$ and a critic $Q_\phi$ using behavioral cloning and FQE (Le et al., 2019), respectively for state-based experiments. We use a slightly different implementation of FQE from the original implementation, where the $\arg\min$ operation is approximated with mini-batch gradient descent, similar to standard Q-learning as shown in Algorithm 2.

---

**Algorithm 2** FQE: Fitted Q Evaluation (Le et al., 2019)

---

**Input:** Offline dataset $\mathcal{D}_{\text{env}}$, policy $\pi_\theta$
1: Randomly initialize Q-function $Q_\phi$
2: **while** not converged **do**
3:    $\{\mathbf{s}_i, \mathbf{a}_i, r_i, \mathbf{s}'_i\}_{i=1}^N \sim \mathcal{D}_{\text{env}}$
4:    $y_i = \text{sg}(r_i + Q_\phi(\mathbf{s}'_i, \pi_\theta(\mathbf{s}'_i)))$             ▷ $\text{sg}(\cdot)$ is stop-gradient operator
5:    $L_{\text{FQE}}(\phi) = \frac{1}{N} \sum_{i=1}^N (Q_\phi(\mathbf{s}_i, \mathbf{a}_i) - y_i)^2$
6:    Update $Q_\phi$ using gradient descent to minimize $L_{\text{FQE}}(\phi)$

---

**Comparisons with prior methods.** We provide a comparison of LEQ with the prior model-based approaches and the baseline methods used in our ablation studies in Table 4.

Table 4: Comparisons with the prior model-based methods and the baseline method. The hyperparameters same with LEQ are colored in **blue**; others are colored in **red**.

| Components | CBOP | MOBILE | MOBILE$^*$ | LEQ (ours) |
|---|---|---|---|---|
| Training scheme | MVE (Feinberg et al., 2018) | MBPO (Janner et al., 2019) | MBPO (Janner et al., 2019) | Dreamer (Hafner et al., 2023) |
| Conservatism | Lower-confidence bound | Lower-confidence bound | Lower-confidence bound | Lower expectile |
| Policy | Stochastic | Stochastic | Stochastic | Deterministic |
| Policy objective | $Q(\mathbf{s}, \mathbf{a})$ | $Q(\mathbf{s}, \mathbf{a})$ | $Q(\mathbf{s}, \mathbf{a})$ | $\lambda$-returns |
| Policy pretraining | BC | – | – | BC |
| # of critics | 20-50 | 2 | 1 | 1 |
| Critic objective | Multi-step (adaptive weighting) | One-step | One-step | $\lambda$-returns |
| Critic pretraining | FQE (Le et al., 2019) | – | – | FQE (Le et al., 2019) |
| Horizon length ($H$) | 10 | 1 | 1 | 10 |
| Expansion length ($R$) | – | 1 or 5 | 10 | 5 |
| Discount rate ($\gamma$) | 0.99 | 0.99 | 0.997 | 0.997 |
| $\beta$ in Equation (7) | 1.0 | 0.95 | 0.25 | 0.25 |
| Impl. tricks | – | Clip Q-values with 0 | LayerNorm + Symlog | LayerNorm + Symlog |
| Running time | 24h | 12h | 40m | 2h |

**Hyperparameters of LEQ.** For state-based experiments, we report task-agnostic hyperparameters of our method in Table 5. We note that **we use the same hyperparameters** across all state-based tasks, except $\tau$. We search the value of $\tau$ in $\{0.1, 0.3, 0.4, 0.5\}$ and report the best value for the main experimental results. In addition, we report the exhaustive results in Tables 15 and 16, and summarize $\tau$ used in the main results in Table 6.

For pixel-based experiments, we decrease the horizon length of DreamerV3 from 15 to 5, since we do not observe performance improvement with the longer imagination horizon (Appendix E), consistent with the finding from Lu et al. (2023). Moreover, we remove the entropy bonus, as exploration is not required in offline RL. We search the value of $\tau$ in $\{0.1, 0.3, 0.4, 0.5\}$ and report the best value in the main experimental results. All other hyperparameters follow the default settings of DreamerV3. We also report the exhaustive results in Table 17, and summarize $\tau$ in Table 9.

Table 5: Shared hyperparameters of LEQ in state-based experiments.

| Hyperparameters | Value | Description |
|---|---|---|
| $lr_{\text{actor}}$ | 3e-5 | Learning rate of actor |
| $lr_{\text{critic}}$ | 1e-4 | Learning rate of critic |
| Optimizer | Adam | Optimizer |
| $T_{\text{expand}}$ | 5000 | Interval of expanding dataset |
| $N_{\text{expand}}$ | 50000 | Number of data for each expansion of dataset |
| $R$ | 5 | Rollout length for dataset expansion |
| $\sigma_{\text{exp}}$ | 1.0 | Exploration noise for dataset expansion |
| $N_{\text{iter}}$ | 1M | Total number of gradient steps. |
| $B_{\text{env}}$ | 256 | Batch size from original dataset |
| $B_{\text{model}}$ | 256 | Batch size from expanded dataset |
| $\gamma$ | 0.997 | Discount factor |
| $\lambda$ | 0.95 | $\lambda$ value for $\lambda$-return |
| $H$ | 10 | Imagination length |
| $\omega_{\text{EMA}}$ | 1 | Weight for critic EMA regularization |
| $\epsilon_{\text{EMA}}$ | 0.995 | Critic EMA decay |

Table 6: Task-specific hyperparameter $\tau$ of LEQ in state-based experiments.

| Domain | Task | $\tau$ |
|---|---|---|
| AntMaze | umaze | 0.1 |
| | umaze-diverse | 0.1 |
| | medium-play | 0.3 |
| | medium-diverse | 0.1 |
| | large-play | 0.3 |
| | large-diverse | 0.3 |
| | ultra-play | 0.1 |
| | ultra-diverse | 0.1 |
| MuJoCo | hopper-r | 0.1 |
| | hopper-m | 0.1 |
| | hopper-mr | 0.3 |
| | hopper-me | 0.1 |
| | walker2d-r | 0.1 |
| | walker2d-m | 0.3 |
| | walker2d-mr | 0.5 |
| | walker2d-me | 0.1 |
| | halfcheetah-r | 0.3 |
| | halfcheetah-m | 0.3 |
| | halfcheetah-mr | 0.4 |
| | halfcheetah-me | 0.1 |
| NeoRL | Hopper-L | 0.1 |
| | Hopper-M | 0.1 |
| | Hopper-H | 0.1 |
| | Walker2d-L | 0.3 |
| | Walker2d-M | 0.1 |
| | Walker2d-H | 0.1 |
| | HalfCheetah-L | 0.1 |
| | HalfCheetah-M | 0.3 |
| | HalfCheetah-H | 0.3 |

**Task-specific hyperparameters of the compared methods.** We report the best hyperparameters of MOBILE* for the AntMaze tasks in Tables 7 and 8. For MOBILE and MOBILE*, we search the value of $c$ within $\{0.1, 0.5, 1.0, 1.5\}$, as suggested in MOBILE (Sun et al., 2023), where $c$ is the coefficient of the penalized bellman operator:

$$T\hat{Q}(\mathbf{s}, \mathbf{a}) = r(\mathbf{s}, \mathbf{a}) + \gamma Q(\mathbf{s}', \mathbf{a}') - c \cdot \text{Std}(Q(\mathbf{s}', \mathbf{a}')). \quad (12)$$

For CBOP, we conduct hyperparameter search for $\psi$ in $\{0.5, 2.0, 3.0, 5.0\}$, as suggested in the original paper, where $\psi$ is an LCB coefficient of CBOP. We do not report the best hyperparameter for MOBILE and CBOP because both methods score zero points for all hyperparameters in AntMaze.

For MOPO in V-D4RL experiments, we search for $\lambda$ in $\{3, 10\}$, as suggested in Lu et al. (2023), where $\lambda$ is the penalization coefficient in MOPO. Then, we report the best value in Table 10. For ROSMO, we use the hyperparameter specified in the official code.

Table 7: Task-specific hyperparameters in MOBILE$^*$.

| Domain | Task | $c$ |
|--------|------|-----|
| AntMaze | umaze | 1.0 |
| | umaze-diverse | 1.0 |
| | medium-play | 1.0 |
| | medium-diverse | 0.1 |
| | large-play | 0.1 |
| | large-diverse | 0.1 |
| | ultra-play | 1.0 |
| | ultra-diverse | 1.0 |

Table 8: Task-specific hyperparameters in MOBILE$^*$ with $\lambda$-returns.

| Domain | Task | $c$ |
|--------|------|-----|
| AntMaze | umaze | 1.0 |
| | umaze-diverse | 0.5 |
| | medium-play | 0.1 |
| | medium-diverse | 0.1 |
| | large-play | 0.1 |
| | large-diverse | 0.1 |
| | ultra-play | 1.0 |
| | ultra-diverse | 0.5 |

Table 9: Task-specific hyperparameter $\tau$ of LEQ in V-D4RL experiments.

| Domain | Task | $\tau$ |
|--------|------|--------|
| walker_walk | random | 0.5 |
| | medium | 0.3 |
| | medium_replay | 0.3 |
| | medium_expert | 0.1 |
| cheetah_run | random | 0.3 |
| | medium | 0.1 |
| | medium_replay | 0.1 |
| | medium_expert | 0.5 |

Table 10: Task-specific hyperparameter $\lambda$ of MOPO in V-D4RL experiments.

| Domain | Task | $\lambda$ |
|--------|------|-----------|
| walker_walk | random | 3.0 |
| | medium | 3.0 |
| | medium_replay | 3.0 |
| | medium_expert | 3.0 |
| cheetah_run | random | 3.0 |
| | medium | 3.0 |
| | medium_replay | 10.0 |
| | medium_expert | 10.0 |

# B    PROOF OF THE POLICY OBJECTIVE

We show that the surrogate loss in Equation (11) leads to a better approximation for the expectile of $\lambda$-returns in Equation (10) than maximizing $Q_\phi(s, a)$. In other words, we show that optimizing the following policy objective:

$$\hat{J}_\lambda(\theta) = \mathbb{E}_{\mathcal{T} \sim p_\psi, \pi_\theta}[(W^\tau(Q_\phi(\mathbf{s}_t, \mathbf{a}_t) > Q_t^\lambda(\mathcal{T}))Q_t^\lambda(\mathcal{T})], \tag{13}$$

leads to optimizing a lower-bias estimator of $\mathbb{E}^\tau_{\mathcal{T} \sim p_\psi, \pi_\theta}[Q_t^\lambda(\mathcal{T})]$ than $Q_\phi(\mathbf{s}_t, \mathbf{a}_t)$.

To show this, we first prove that $\hat{Y}_{\text{new}} = \frac{\mathbb{E}[W^\tau(Q_\phi(\mathbf{s}_t, \mathbf{a}_t) > Q_t^\lambda(\mathcal{T})) \cdot Q_t^\lambda(\mathcal{T})]}{\mathbb{E}[W^\tau(Q_\phi(\mathbf{s}_t, \mathbf{a}_t) > Q_t^\lambda(\mathcal{T}))]}$ is closer to $\mathbb{E}^\tau_{\mathcal{T} \sim p_\psi, \pi_\theta}[Q_t^\lambda(\mathcal{T})]$ than $Q_\phi(\mathbf{s}_t, \mathbf{a}_t)$ for *most of the situations*. For deriving the proof, we generalize the problem by considering an arbitrary distribution $X$ and its estimate $\hat{Y}$, which correspond to $X = Q_t^\lambda(\tau), \hat{Y} = Q_\phi(\mathbf{s}, \mathbf{a})$. Then, we show $\hat{Y}_{\text{new}} = \frac{\mathbb{E}[W^\tau(\hat{Y} > X) \cdot X]}{\mathbb{E}[W^\tau(\hat{Y} > X)]}$ is closer to $Y$ than $\hat{Y}$. We split the case to two cases where $\hat{Y} \geq Y$ (Lemma 1) and $\hat{Y} \leq Y$ (Lemma 2) and show the effectiveness of $\hat{Y}_{\text{new}}$ instead of $\hat{Y}$ for each case.

**Lemma 1.** *Let $X$ be a distribution and $Y = E^\tau[X]$ be a lower expectile of $X$ (i.e. $0 < \tau \leq 0.5$). Let $\hat{Y}$ be an arbitrary **optimistic** estimate of $Y$ (i.e., $\hat{Y} \geq Y$), and define $W^\tau(\cdot) = |\tau - \mathbb{1}(\cdot)|$. If we let $\hat{Y}_{new} = \frac{\mathbb{E}[W^\tau(\hat{Y} > X) \cdot X]}{\mathbb{E}[W^\tau(\hat{Y} > X)]}$ be a new estimate of $Y$, then $|\hat{Y}_{new} - Y| \leq |\hat{Y} - Y|$.*

*Proof.*

$|\hat{Y}_{\text{new}} - Y|$

$$= \left| \frac{\mathbb{E}[W^\tau(\hat{Y} > X) \cdot X]}{\mathbb{E}[W^\tau(\hat{Y} > X)]} - \frac{\mathbb{E}[W^\tau(Y > X) \cdot X]}{\mathbb{E}[W^\tau(Y > X)]} \right| \qquad (\because \text{Def. of } \hat{Y}_{\text{new}} \text{ and } Y)$$

$$= \left| \frac{\mathbb{E}[W^\tau(Y > X) \cdot X] + \mathbb{E}[(1 - 2\tau) \cdot \mathbb{1}(Y \leq X \leq \hat{Y}) \cdot X]}{\mathbb{E}[W^\tau(Y > X)] + \mathbb{E}[(1 - 2\tau) \cdot \mathbb{1}(Y \leq X \leq \hat{Y})]} - \frac{\mathbb{E}[W^\tau(Y > X) \cdot X]}{\mathbb{E}[W^\tau(Y > X)]} \right| \qquad (\because \text{Def. of } W^\tau(\cdot))$$

$$= \left| \frac{\mathbb{E}[W^\tau(Y > X)]\mathbb{E}[(1 - 2\tau) \cdot \mathbb{1}(Y \leq X \leq \hat{Y}) \cdot X] - \mathbb{E}[W^\tau(Y > X) \cdot X]\mathbb{E}[(1 - 2\tau) \cdot \mathbb{1}(Y \leq X \leq \hat{Y})]}{\mathbb{E}[W^\tau(Y > X)](\mathbb{E}[W^\tau(Y > X)] + \mathbb{E}[(1 - 2\tau) \cdot \mathbb{1}(Y \leq X \leq \hat{Y})])} \right|$$

$$= (1 - 2\tau) \cdot \left| \frac{\mathbb{E}[W^\tau(Y > X)]\mathbb{E}[\mathbb{1}(Y \leq X \leq \hat{Y}) \cdot X] - \mathbb{E}[W^\tau(Y > X) \cdot X]\mathbb{E}[\mathbb{1}(Y \leq X \leq \hat{Y})]}{\mathbb{E}[W^\tau(Y > X)](\mathbb{E}[W^\tau(Y > X)] + \mathbb{E}[(1 - 2\tau) \cdot \mathbb{1}(Y \leq X \leq \hat{Y})])} \right|$$

$$= (1 - 2\tau) \cdot \left| \frac{\mathbb{E}[\mathbb{1}(Y \leq X \leq \hat{Y}) \cdot X] - Y p(Y \leq X \leq \hat{Y})}{\mathbb{E}[W^\tau(Y > X)] + \mathbb{E}[(1 - 2\tau) \cdot \mathbb{1}(Y \leq X \leq \hat{Y})]} \right|$$

$$= (1 - 2\tau) \cdot \left| \frac{p(Y \leq X \leq \hat{Y})(\mathbb{E}_{Y \leq X \leq \hat{Y}}[X] - Y)}{\mathbb{E}[W^\tau(Y > X)] + \mathbb{E}[(1 - 2\tau) \cdot \mathbb{1}(Y \leq X \leq \hat{Y})]} \right|$$

$$= (1 - 2\tau) \cdot \left| \frac{p(Y \leq X \leq \hat{Y})(\mathbb{E}_{Y \leq X \leq \hat{Y}}[X] - Y)}{(1 - \tau) - (1 - 2\tau) \cdot p(Y \leq X) + (1 - 2\tau) \cdot p(Y \leq X \leq \hat{Y})} \right|$$

$$= (1 - 2\tau) \cdot \left| \frac{p(Y \leq X \leq \hat{Y})(\mathbb{E}_{Y \leq X \leq \hat{Y}}[X] - Y)}{(1 - \tau) - (1 - 2\tau) \cdot p(\hat{Y} \leq X)} \right|$$

$$= (1 - 2\tau) \cdot \left| \frac{p(Y \leq X \leq \hat{Y})(\mathbb{E}_{Y \leq X \leq \hat{Y}}[X] - Y)}{\tau + (1 - 2\tau) \cdot p(X \leq \hat{Y})} \right|$$

$$\leq \frac{p(Y \leq X \leq \hat{Y})(\mathbb{E}_{Y \leq X \leq \hat{Y}}[X] - Y)}{p(X \leq \hat{Y})}$$

$$\leq \mathbb{E}_{Y \leq X \leq \hat{Y}}[X] - Y$$

$$\leq \hat{Y} - Y = |\hat{Y} - Y|$$

$\square$

Lemma 1 shows that when the estimated value is optimistic ($\hat{Y} \geq Y$), the bias of the new estimate is always smaller than that of the original estimate. In the context of LEQ algorithm, the lemma tells that if the critic network ($\hat{Y}$) overestimates the lower-expectile of the target returns ($Y = \mathbb{E}^\tau[X]$), the surrogate loss ($Y_{\text{new}}$) compensates the overestimation of the critic values.

Unfortunately, when the estimated value is pessimistic ($\hat{Y} \leq Y$), there are some exceptional cases that the surrogate loss overcompensates the underestimation, resulting in an even larger error. The boundary for these cases is characterized in Lemma 2:

**Lemma 2.** *Let $X$ be a distribution and $Y = E^\tau[X]$ be a lower expectile of $X$ (i.e. $0 < \tau \leq 0.5$). Let $\hat{Y}$ be an arbitrary **conservative** estimate of $Y$ (i.e., $\hat{Y} \leq Y$), and define $W^\tau(\cdot) = |\tau - \mathbb{1}(\cdot)|$. If we let $\hat{Y}_{new} = \frac{\mathbb{E}[W^\tau(\hat{Y}>X)\cdot X]}{\mathbb{E}[W^\tau(\hat{Y}>X)]}$ be a new estimate of $Y$, then $|\hat{Y}_{new} - Y| \leq |\hat{Y} - Y|$, when $p(X \leq \hat{Y}) \geq \frac{1}{2}(p(X \leq Y) - \frac{\tau}{1-2\tau})$.*

*Proof.*

$|\hat{Y}_{\text{new}} - Y|$

$$= \left| \frac{\mathbb{E}[W^\tau(\hat{Y} > X) \cdot X]}{\mathbb{E}[W^\tau(\hat{Y} > X)]} - \frac{\mathbb{E}[W^\tau(Y > X) \cdot X]}{\mathbb{E}[W^\tau(Y > X)]} \right| \qquad (\because \text{Def. of } \hat{Y}_{\text{new}} \text{ and } Y)$$

$$= \left| \frac{\mathbb{E}[W^\tau(Y > X) \cdot X] - \mathbb{E}[(1-2\tau) \cdot \mathbb{1}(\hat{Y} \leq X \leq Y)X]}{\mathbb{E}[W^\tau(Y > X)] - \mathbb{E}[(1-2\tau) \cdot \mathbb{1}(\hat{Y} \leq X \leq Y)]} - \frac{\mathbb{E}[W^\tau(Y > X) \cdot X]}{\mathbb{E}[W^\tau(Y > X)]} \right| \quad (\because \text{Def. of } W^\tau(\cdot))$$

$$= \left| \frac{\mathbb{E}[W^\tau(Y > X)]\mathbb{E}[(1-2\tau) \cdot \mathbb{1}(\hat{Y} \leq X \leq Y) \cdot X] - \mathbb{E}[W^\tau(Y > X) \cdot X]\mathbb{E}[(1-2\tau) \cdot \mathbb{1}(\hat{Y} \leq X \leq Y)]}{\mathbb{E}[W^\tau(Y > X)](\mathbb{E}[W^\tau(Y > X)] - \mathbb{E}[(1-2\tau) \cdot \mathbb{1}(\hat{Y} \leq X \leq Y)])} \right|$$

$$= (1-2\tau) \cdot \left| \frac{\mathbb{E}[W^\tau(Y > X)X]\mathbb{E}[\mathbb{1}(\hat{Y} \leq X \leq Y)] - \mathbb{E}[W^\tau(Y > X)]\mathbb{E}[\mathbb{1}(\hat{Y} \leq X \leq Y) \cdot X]}{\mathbb{E}[W^\tau(Y > X)](\mathbb{E}[W^\tau(Y > X)] - \mathbb{E}[(1-2\tau) \cdot \mathbb{1}(\hat{Y} \leq X \leq Y)])} \right|$$

$$= (1-2\tau) \cdot \left| \frac{Yp(\hat{Y} \leq X \leq Y) - \mathbb{E}[\mathbb{1}(\hat{Y} \leq X \leq Y) \cdot X]}{\mathbb{E}[W^\tau(Y > X)] - \mathbb{E}[(1-2\tau) \cdot \mathbb{1}(\hat{Y} \leq X \leq Y)]} \right|$$

$$= (1-2\tau) \cdot \left| \frac{p(\hat{Y} \leq X \leq Y)(Y - \mathbb{E}_{\hat{Y} \leq X \leq Y}[X])}{\mathbb{E}[W^\tau(Y > X)] - \mathbb{E}[(1-2\tau) \cdot \mathbb{1}(\hat{Y} \leq X \leq Y)]} \right|$$

$$= (1-2\tau) \cdot \left| \frac{p(\hat{Y} \leq X \leq Y)(Y - \mathbb{E}_{\hat{Y} \leq X \leq Y}[X])}{(1-\tau) - (1-2\tau) \cdot p(Y \leq X) - (1-2\tau) \cdot p(\hat{Y} \leq X \leq Y)} \right|$$

$$= (1-2\tau) \cdot \left| \frac{p(\hat{Y} \leq X \leq Y)(Y - \mathbb{E}_{\hat{Y} \leq X \leq Y}[X])}{(1-\tau) - (1-2\tau) \cdot p(\hat{Y} \leq X)} \right|$$

$$= (1-2\tau) \cdot \left| \frac{p(\hat{Y} \leq X \leq Y)(Y - \mathbb{E}_{\hat{Y} \leq X \leq Y}[X])}{\tau + (1-2\tau) \cdot p(X \leq \hat{Y})} \right|$$

$$= \left| \frac{p(\hat{Y} \leq X \leq Y)(Y - \mathbb{E}_{\hat{Y} \leq X \leq Y}[X])}{\frac{\tau}{1-2\tau} + p(X \leq \hat{Y})} \right|$$

$$= \frac{p(X \leq Y) - p(X \leq \hat{Y})}{\frac{\tau}{1-2\tau} + p(X \leq \hat{Y})} \cdot (Y - \mathbb{E}_{\hat{Y} \leq X \leq Y}[X])$$

$$\leq Y - \mathbb{E}_{\hat{Y} \leq X \leq Y}[X]$$

$$\leq Y - \hat{Y} = |\hat{Y} - Y|$$

$\square$

By combining Lemma 1, Lemma 2, we get Theorem 1 as below:

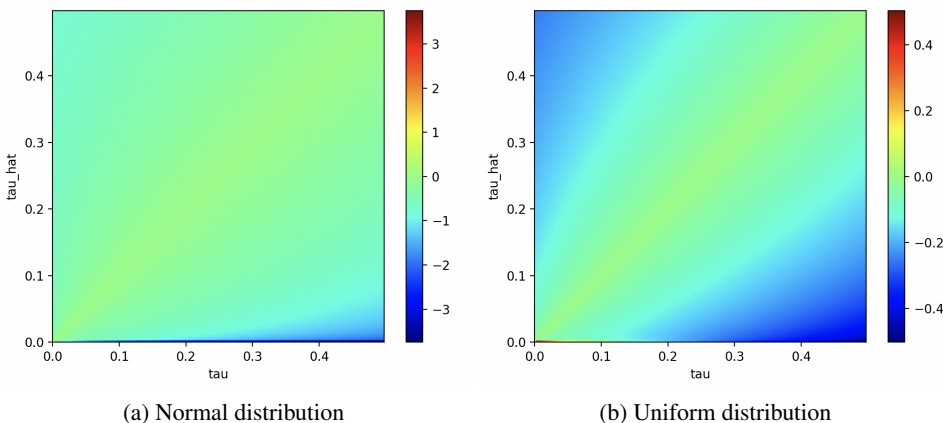

(a) Normal distribution              (b) Uniform distribution

Figure 11: **Region of exceptions for the surrogate loss in realistic distributions.** We visualize the cases where the surrogate loss is worse than the direct policy optimization, for two common distributions for $X$: (a) a normal distribution $\mathcal{N}(0, 1)$ and (b) a uniform distribution $\mathcal{U}(0, 1)$. We sample $\tau$ and $\hat{\tau}$ from `arange`$(0, 0.5, 0.0025)$ and compute $Y = \mathbb{E}^\tau[X]$, $\hat{Y} = \mathbb{E}^{\hat\tau}[X]$. The plots show the difference of errors $|\hat{Y}_{\text{new}} - Y| - |\hat{Y} - Y|$: positive values (red to yellow) indicate that the surrogate loss performs worse than the estimate, while negative values (green to blue) indicate that the surrogate loss outperforms the estimate. There is no exception for normal distribution, and only 39 out of 40000 cases are exceptional for uniform distribution.

**Theorem 1.** *Let $X$ be a distribution and $Y = E^\tau[X]$ be a lower expectile of $X$ (i.e. $0 < \tau \leq 0.5$). Let $\hat{Y}$ be an arbitrary estimate of $Y$, and define $W^\tau(\cdot) = |\tau - \mathbb{1}(\cdot)|$. If we let $\hat{Y}_{new} = \frac{\mathbb{E}[W^\tau(\hat{Y} > X) \cdot X]}{\mathbb{E}[W^\tau(\hat{Y} > X)]}$ be a new estimate of $Y$, then $|\hat{Y}_{new} - Y| \leq |\hat{Y} - Y|$ if the following condition holds:*

$$p(X \leq \hat{Y}) \geq \frac{1}{2}(p(X \leq Y) - \frac{\tau}{1 - 2\tau}). \tag{14}$$

*Proof.* We combine the two cases dealt in Lemma 1, Lemma 2.

Here, $\hat{Y} \geq Y$ from Lemma 1 can be omitted, since the condition of Equation (14) includes the case of $\hat{Y} \geq Y$.

$\because$ if $\hat{Y} \geq Y$, then $p(X \leq \hat{Y}) \geq p(X \leq Y) \geq \frac{1}{2}(p(X \leq Y) - \frac{\tau}{1 - 2\tau})$. $\qquad\qquad\square$

The condition of Equation (14) is a conservative bound applicable to any distribution, and $|\hat{Y}_{\text{new}} - Y| \leq |\hat{Y} - Y|$ holds across much broader regions in general. For example, Figure 11 shows that the inequality holds for 100%, 99.9% of the cases for normal, uniform distributions, respectively.

Here, we illustrate how we optimize $\mathbb{E}[W^\tau(Q_\phi(\mathbf{s}_t, \mathbf{a}_t) > Q_t^\lambda(\mathcal{T})) \cdot Q_t^\lambda(\mathcal{T})]$ instead of $\frac{\mathbb{E}[W^\tau(Q_\phi(\mathbf{s}_t, \mathbf{a}_t) > Q_t^\lambda(\mathcal{T})) \cdot Q_t^\lambda(\mathcal{T})]}{\mathbb{E}[W^\tau(Q_\phi(\mathbf{s}_t, \mathbf{a}_t) > Q_t^\lambda(\mathcal{T}))]}$. The normalizing factor $\mathbb{E}[W^\tau(Q_\phi(\mathbf{s}_t, \mathbf{a}_t) > Q_t^\lambda(\mathcal{T}))]$ is non-differentiable with $\mathcal{T}$ and the gradient is 0 everywhere (except $Q_\phi(\mathbf{s}_t, \mathbf{a}_t) = Q_t^\lambda(\mathcal{T})$). Thus, if we calculate the gradient of $\hat{Y}_{\text{new}}$, the gradient for the normalizing factor disappears. Therefore, we omit the normalizing factor and get an equivalent formula $\mathbb{E}[W^\tau(Q_\phi(\mathbf{s}_t, \mathbf{a}_t) > Q_t^\lambda(\mathcal{T})) \cdot Q_t^\lambda(\mathcal{T})]$ for gradient-based optimization.

## C  COMPLETE NUMERICAL RESULTS

For completeness, we provide the full tabular results corresponding to the graphical summaries presented in Figures 5, 6, 8 and 9. Some low-performing baselines were omitted from the graphical summaries for clarity but are included in the table below for reference.

Table 11: **AntMaze results.** Each number represents the average success rate on 100 trials over different seeds. The results for **LEQ**, **MOBILE**, and **CBOP** are averaged over 5 seeds. The results for other methods are reported following their respective papers.

| | Model-free | | Seq. modeling | | Model-based | | | | | |
|---|---|---|---|---|---|---|---|---|---|---|
| Dataset | CQL | IQL | TT | TAP | COMBO | RAMBO | MOBILE† | CBOP† | IQL-TD-MPC | LEQ (ours) |
| umaze | 74.0 | 87.5 | **100.0** | 81.5 | 80.3 | 25.0 | 0.0 ±0.0 | 0.0 ±0.0 | 52.0 | 94.4 ±6.3 |
| umaze-diverse | **84.0** | 62.2 | 21.5 | 68.5 | 57.3 | 0.0 | 0.0 ±0.0 | 0.0 ±0.0 | 72.6 | 71.0 ±12.3 |
| medium-play | 61.2 | 71.2 | **93.3** | 78.0 | 0.0 | 16.4 | 0.0 ±0.0 | 0.0 ±0.0 | **88.8** | 58.8 ±33.0 |
| medium-diverse | 53.7 | **70.0** | **100.0** | 85.0 | 0.0 | 23.2 | 0.0 ±0.0 | 0.0 ±0.0 | 40.3 | 46.2 ±23.2 |
| large-play | 15.8 | 39.6 | 66.7 | **74.0** | 0.0 | 0.0 | 0.0 ±0.0 | 0.0 ±0.0 | 66.6 | 58.6 ±9.1 |
| large-diverse | 14.9 | 47.5 | 60.0 | **82.0** | 0.0 | 2.4 | 0.0 ±0.0 | 0.0 ±0.0 | 4.0 | 60.2 ±18.3 |
| ultra-play | — | 8.3 | 20.0 | 22.0 | — | — | 0.0 ±0.0 | 0.0 ±0.0 | 20.6 | **25.8 ±18.2** |
| ultra-diverse | — | 15.6 | 33.3 | 26.0 | — | — | 0.0 ±0.0 | 0.0 ±0.0 | 3.6 | **55.8 ±18.3** |
| Total w/o **ultra** | 303.6 | 354.1 | 441.5 | **469.0** | 137.6 | 67.0 | 0.0 | 0.0 | 324.3 | 388.8 |
| Total | — | 378.0 | **494.8** | **517.0** | — | — | 0.0 | 0.0 | 348.5 | 470.4 |

†We use the official implementation of **MOBILE** and **CBOP**.

Table 12: **NeoRL results. LEQ** and **IQL** results are averaged over 5 seeds. The results for prior works are reported following Sun et al. (2023) and Qin et al. (2022). **MOPO**\* is an improved version of **MOPO** presented in Sun et al. (2023). We highlight the results better than 95% of the best score.

| | Model-free | | | | | Model-based | | |
|---|---|---|---|---|---|---|---|---|
| Dataset | BC | TD3+BC | CQL | EDAC | IQL | MOPO* | MOBILE | LEQ (ours) |
| Hopper-L | 15.1 | 15.8 | 16.0 | 18.3 | 16.7 | 6.2 | 17.4 | **24.2** ±2.3 |
| Hopper-M | 51.3 | 70.3 | 64.5 | 44.9 | 28.4 | 1.0 | 51.1 | **104.3** ±5.2 |
| Hopper-H | 43.1 | 75.3 | 76.6 | 52.5 | 22.3 | 11.5 | 87.8 | **95.5** ±13.9 |
| Walker2d-L | 28.5 | 43.0 | 44.7 | 40.2 | 30.7 | 11.6 | 37.6 | **65.1** ±2.3 |
| Walker2d-M | 48.7 | 58.5 | 57.3 | 57.6 | 51.8 | 39.9 | **62.2** | 45.2 ±19.4 |
| Walker2d-H | **72.6** | 69.6 | **75.3** | **75.5** | **76.3** | 18.0 | **74.9** | 73.7 ±1.1 |
| HalfCheetah-L | 29.1 | 30.0 | 38.2 | 31.3 | 30.7 | 40.1 | **54.7** | 33.4 ±1.6 |
| HalfCheetah-M | 49.0 | 52.3 | 54.6 | 54.9 | 51.8 | 62.3 | **77.8** | 59.2 ±3.9 |
| HalfCheetah-H | 71.4 | 75.3 | 77.4 | **81.4** | 76.3 | 65.9 | **83.0** | 71.8 ±8.0 |
| **Total** | 408.8 | 490.1 | 504.6 | 456.6 | 385.0 | 256.5 | **546.5** | **572.4** |

Table 13: **D4RL MuJoCo Gym results.** Each number is a normalized score averaged over 100 trials (Fu et al., 2020). Our results are averaged over 5 seeds. The results for prior works are reported following their respective papers. **MOPO**\* is an improved version of **MOPO**, introduced in Sun et al. (2023). We highlight the results that are better than 95% of the best score.

| | Model-free | | | Seq. modeling | | Model-based | | | | | |
|---|---|---|---|---|---|---|---|---|---|---|---|
| Dataset | CQL | EDAC | IQL | TT | TAP | MOPO* | COMBO | RAMBO | MOBILE | CBOP | LEQ (ours) |
| hopper-r | 5.3 | 25.3 | 7.6 | 6.9 | - | 31.7 | 17.9 | 25.4 | **31.9** | 32.8 | 32.4 ±0.3 |
| hopper-m | 61.9 | **101.6** | 66.3 | 67.4 | 63.4 | 62.8 | 97.2 | 87.0 | **106.6** | 102.6 | 103.4 ±0.3 |
| hopper-mr | 86.3 | **101.0** | 94.7 | 99.4 | 87.3 | **99.4** | **103.5** | 89.5 | 99.5 | 104.3 | 103.9 ±1.3 |
| hopper-me | 96.9 | **110.7** | 91.5 | **110.0** | 105.5 | 81.6 | **111.1** | 88.2 | **112.6** | 111.6 | 109.4 ±1.8 |
| walker2d-r | 5.4 | 16.6 | 5.2 | 5.9 | - | 7.4 | 7.0 | 0.0 | 17.9 | 17.8 | **21.5** ±0.1 |
| walker2d-m | 79.5 | **92.5** | 78.3 | 84.9 | 64.9 | 81.3 | 84.1 | 81.9 | 84.9 | **87.7** | 74.9 ±26.9 |
| walker2d-mr | 76.8 | 87.1 | 73.9 | 89.2 | 66.8 | 85.6 | 56.0 | 89.2 | 89.9 | 92.7 | **98.7** ±6.0 |
| walker2d-me | 109.1 | **114.7** | 109.6 | 101.9 | 107.4 | **112.9** | 103.3 | 56.7 | **115.2** | 117.2 | 108.2 ±1.3 |
| halfcheetah-r | 31.3 | 28.4 | 11.8 | 6.1 | - | **38.5** | **38.8** | **39.5** | **39.3** | 32.8 | 30.8 ±3.3 |
| halfcheetah-m | 46.9 | 65.9 | 47.4 | 46.9 | 45.0 | 73.0 | 54.2 | **77.9** | **74.6** | **74.3** | 71.7 ±4.4 |
| halfcheetah-mr | 45.3 | 61.3 | 44.2 | 44.1 | 40.8 | **72.1** | 55.1 | 68.7 | 71.7 | 66.4 | 65.5 ±1.1 |
| halfcheetah-me | 95.0 | **106.3** | 86.7 | 95.0 | 91.8 | 90.8 | 90.0 | 95.4 | **108.2** | 105.4 | 102.8 ±0.4 |
| **Total** | 739.7 | 911.4 | 717.2 | 747.5 | - | 844.0 | 802.0 | 812.4 | **959.5** | 953.4 | 923.2 |

Table 14: **V-D4RL results.** We report the mean and standard deviation of returns over 3 seeds. For **ROSMO**, we replace the categorical distribution in their official code with Gaussian distribution to support continuous action spaces. For **MOPO**, we implement **MOPO** on top of DreamerV3. The results for other prior works are reported following Lu et al. (2023); Islam et al. (2023).

| | Model-free | | | | Model-based | | | |
|---|---|---|---|---|---|---|---|---|
| **Method** | **BC** | **CQL** | **DrQ+BC** | **ACRO** | **OfflineDV2** | **MOPO** | **ROSMO** | **LEQ** |
| walker_walk-random | 2.0 | 14.4 | 5.5 | 0.0 | **28.7** | $3.2_{\pm0.4}$ | $2.9_{\pm0.4}$ | $22.4_{\pm1.1}$ |
| walker_walk-medium | 40.9 | 14.8 | **46.8** | **48.7** | 34.1 | $37.1_{\pm3.7}$ | $49.8_{\pm2.3}$ | $43.1_{\pm3.2}$ |
| walker_walk-medium_replay | 16.5 | 11.4 | 28.7 | 27.8 | **56.5** | $11.4_{\pm9.2}$ | $28.1_{\pm0.9}$ | $43.0_{\pm7.3}$ |
| walker_walk-medium_expert | 47.7 | 56.4 | **86.4** | **91.4** | 43.9 | $46.6_{\pm3.5}$ | $82.9_{\pm0.7}$ | $87.2_{\pm2.4}$ |
| cheetah_run-random | 0.0 | 5.9 | 5.8 | 0.0 | **31.7** | $3.2_{\pm3.9}$ | $2.5_{\pm0.5}$ | $14.8_{\pm1.0}$ |
| cheetah_run-medium | **51.6** | 40.9 | **53.0** | 52.8 | 17.2 | $34.7_{\pm6.6}$ | $50.8_{\pm2.5}$ | $37.9_{\pm8.0}$ |
| cheetah_run-medium_replay | 25.0 | 10.7 | 44.8 | 41.7 | **61.5** | $30.3_{\pm5.1}$ | $48.6_{\pm1.5}$ | $34.3_{\pm1.1}$ |
| cheetah_run-medium_expert | **57.5** | 20.9 | 50.6 | 46.6 | 10.4 | $36.6_{\pm5.6}$ | $45.4_{\pm2.2}$ | $25.1_{\pm6.6}$ |
| Total | 241.2 | 175.4 | **321.6** | **309.0** | 284.0 | 203.3 | **310.9** | **307.6** |

# D    RESULTS FOR ALL EXPECTILES $\tau$

To give insights how the expectile parameter $\tau$ affects the performance of LEQ, we report the performance of LEQ with all expectile values $\{0.1, 0.3, 0.4, 0.5\}$. The expectile parameter $\tau$ has a trade-off – high expectile makes the model's predictions less conservative while making a policy easily exploit the model. We recommend first trying $\tau = 0.1$, which works well for most of the tasks, and increase $\tau$ until the performance starts to drop.

Table 15: **Antmaze results of LEQ with different expectiles.** We report the results in Antmaze task with expectiles value of 0.1, 0.3, 0.4, 0.5. The best value is highlighted.

| Expectile | 0.1 | 0.3 | 0.4 | 0.5 |
|---|---|---|---|---|
| antmaze-umaze | **94.4** $_{\pm 6.3}$ | 39.0 $_{\pm 28.1}$ | 0.2 $_{\pm 0.4}$ | 3.0 $_{\pm 5.5}$ |
| antmaze-umaze-diverse | **71.0** $_{\pm 12.2}$ | 23.6 $_{\pm 21.7}$ | 4.0 $_{\pm 4.2}$ | 0.0 $_{\pm 0.0}$ |
| antmaze-medium-play | 50.2 $_{\pm 39.9}$ | **58.8** $_{\pm 33.0}$ | 36.0 $_{\pm 21.8}$ | 0.6 $_{\pm 1.2}$ |
| antmaze-medium-diverse | **46.2** $_{\pm 23.2}$ | 13.2 $_{\pm 13.3}$ | 11.6 $_{\pm 14.8}$ | 10.6 $_{\pm 13.3}$ |
| antmaze-large-play | 42.0 $_{\pm 30.6}$ | **58.6** $_{\pm 9.1}$ | 52.2 $_{\pm 15.8}$ | 42.2 $_{\pm 7.3}$ |
| antmaze-large-diverse | **60.6** $_{\pm 32.1}$ | 60.2 $_{\pm 18.3}$ | 48.8 $_{\pm 5.8}$ | 36.8 $_{\pm 9.7}$ |
| antmaze-ultra-play | **25.8** $_{\pm 18.2}$ | 10.8$_{\pm 8.8}$ | 11.6 $_{\pm 12.5}$ | 9.2 $_{\pm 11.5}$ |
| antmaze-ultra-diverse | **55.8** $_{\pm 18.3}$ | 4.6$_{\pm 3.4}$ | 7.6 $_{\pm 7.3}$ | 0.6 $_{\pm 1.2}$ |

Table 16: **D4RL mujoco results of LEQ with different expectiles.** We report the results in D4RL mujoco task with expectiles value of 0.1, 0.3, 0.4, 0.5. The best value is highlighted.

| Expectile | 0.1 | 0.3 | 0.4 | 0.5 |
|---|---|---|---|---|
| hopper-r | **32.4** $_{\pm 0.3}$ | 13.7 $_{\pm 9.1}$ | 16.4 $_{\pm 9.3}$ | 12.5 $_{\pm 10.1}$ |
| hopper-m | **103.4** $_{\pm 0.3}$ | 102.7 $_{\pm 1.7}$ | 81.4 $_{\pm 24.8}$ | 38.6 $_{\pm 29.2}$ |
| hopper-mr | 103.2 $_{\pm 1.0}$ | **103.9** $_{\pm 1.3}$ | 71.5 $_{\pm 34.7}$ | 103.8 $_{\pm 1.9}$ |
| hopper-me | **109.4** $_{\pm 1.8}$ | 108.0 $_{\pm 8.7}$ | 64.2 $_{\pm 35.8}$ | 33.7 $_{\pm 0.5}$ |
| walker2d-r | **21.5** $_{\pm 0.1}$ | 21.5 $_{\pm 0.5}$ | 14.0 $_{\pm 8.8}$ | 8.7 $_{\pm 6.7}$ |
| walker2d-m | 26.3 $_{\pm 37.4}$ | **74.9** $_{\pm 26.9}$ | 60.3 $_{\pm 40.9}$ | 34.8 $_{\pm 34.3}$ |
| walker2d-mr | 48.6 $_{\pm 19.5}$ | 60.5 $_{\pm 27.4}$ | 88.5 $_{\pm 3.5}$ | **98.7** $_{\pm 6.0}$ |
| walker2d-me | **108.2** $_{\pm 1.3}$ | 98.8 $_{\pm 28.8}$ | 105.8 $_{\pm 25.9}$ | 33.7 $_{\pm 31.9}$ |
| halfcheetah-r | 23.8 $_{\pm 1.8}$ | **30.8** $_{\pm 3.3}$ | 29.0 $_{\pm 2.9}$ | 30.2 $_{\pm 2.5}$ |
| halfcheetah-m | 65.3 $_{\pm 2.0}$ | **71.7** $_{\pm 4.4}$ | 58.5 $_{\pm 23.8}$ | 55.5 $_{\pm 16.7}$ |
| halfcheetah-mr | 60.6 $_{\pm 1.4}$ | 55.4 $_{\pm 27.3}$ | **65.5** $_{\pm 1.1}$ | 52.4 $_{\pm 26.7}$ |
| halfcheetah-me | **102.8** $_{\pm 0.4}$ | 81.5 $_{\pm 19.6}$ | 58.1 $_{\pm 26.1}$ | 46.3 $_{\pm 17.7}$ |

Table 17: **V-D4RL results of LEQ with different expectiles.** We report the results in Antmaze task with expectiles value of 0.1, 0.3, 0.4, 0.5. The best value is highlighted.

| Expectile | 0.1 | 0.3 | 0.4 | 0.5 |
|---|---|---|---|---|
| walker_walk-random | 14.5 $_{\pm 1.1}$ | 20.2 $_{\pm 3.6}$ | **22.4** $_{\pm 1.1}$ | 21.8 $_{\pm 0.8}$ |
| walker_walk-medium | **43.1** $_{\pm 3.2}$ | 37.2 $_{\pm 5.3}$ | 35.0 $_{\pm 3.7}$ | 26.6 $_{\pm 1.9}$ |
| walker_walk-medium_replay | 40.6 $_{\pm 6.6}$ | 40.8 $_{\pm 3.4}$ | 41.3 $_{\pm 6.3}$ | **43.0** $_{\pm 7.3}$ |
| walker_walk-medium_expert | **87.2** $_{\pm 2.4}$ | 82.7 $_{\pm 5.0}$ | 77.0 $_{\pm 9.2}$ | 85.0 $_{\pm 1.6}$ |
| cheetah_run-random | 12.1 $_{\pm 1.9}$ | **14.8** $_{\pm 1.0}$ | 14.2 $_{\pm 1.6}$ | 14.6 $_{\pm 2.9}$ |
| cheetah_run-medium | 25.0 $_{\pm 5.9}$ | **37.9** $_{\pm 8.0}$ | 23.6 $_{\pm 11.9}$ | 32.2 $_{\pm 6.2}$ |
| cheetah_run-medium_replay | **34.3** $_{\pm 1.1}$ | 32.3 $_{\pm 2.1}$ | 31.2 $_{\pm 3.5}$ | 31.2 $_{\pm 0.4}$ |
| cheetah_run-medium_expert | 23.9 $_{\pm 4.7}$ | 18.9 $_{\pm 3.9}$ | **25.1** $_{\pm 6.6}$ | 24.3 $_{\pm 7.6}$ |
| Total | 280.7 | 284.8 | 269.9 | 278.6 |

# E    MORE ABLATION RESULTS

**Ablation of each component in LEQ.**    In Table 18, we ablate LEQ's design choices on AntMaze tasks to investigate their importance. Notably, the performance of LEQ drops significantly if we remove the Q-learning loss (461.8 to 312.8) or apply expectile regression for real transitions (461.8 to 249.2), showing the importance of utilizing the real transitions in long-horizon tasks. Moreover, the expectile loss from the policy learning is crucial (461.8 to 357.0), since simply maximizing $\lambda$-returns, even if $Q(\mathbf{s}, \mathbf{a})$ is conservative, can lead to a suboptimal policy due to the lack of conservatism in the rewards from model rollouts. Minor design choices, such as EMA regularization and dataset expansion, have a small impact on the performance in AntMaze.

Table 18: **Ablation studies about various components in LEQ on AntMaze.** We conduct ablation studies on (1) EMA regularization, (2) dataset expansion, (3) utilizing data transitions, (4) lower expectile policy optimization, (5) lower expectile regression for data transitions

| Method | umaze | | medium | | large | | ultra | | Total |
|---|---|---|---|---|---|---|---|---|---|
| | umaze | diverse | play | diverse | play | diverse | play | diverse | |
| LEQ | **94.4** $_{\pm 6.3}$ | 71.0 $_{\pm 12.3}$ | 50.2 $_{\pm 39.9}$ | **46.2** $_{\pm 23.2}$ | 58.6 $_{\pm 9.1}$ | 60.2 $_{\pm 18.3}$ | 25.8 $_{\pm 18.2}$ | **55.8** $_{\pm 18.3}$ | **461.8** |
| - EMA regularization | **96.0** $_{\pm 4.0}$ | 65.6 $_{\pm 5.1}$ | 33.8 $_{\pm 31.4}$ | 21.2 $_{\pm 26.1}$ | 58.8 $_{\pm 8.2}$ | 62.2 $_{\pm 15.0}$ | 8.0 $_{\pm 13.9}$ | 33.5 $_{\pm 35.1}$ | 379.1 |
| - dataset expansion | **97.4** $_{\pm 1.4}$ | 63.0 $_{\pm 23.2}$ | 58.2 $_{\pm 28.0}$ | 28.6 $_{\pm 33.7}$ | 56.0 $_{\pm 9.8}$ | 57.0 $_{\pm 4.5}$ | **39.2** $_{\pm 15.1}$ | 36.0 $_{\pm 12.0}$ | 435.4 |
| - data transitions | 63.0 $_{\pm 19.8}$ | **75.3** $_{\pm 4.2}$ | 31.8 $_{\pm 34.0}$ | 28.0 $_{\pm 27.0}$ | 50.5 $_{\pm 20.9}$ | 57.5 $_{\pm 8.6}$ | 0.8 $_{\pm 1.3}$ | 6.0 $_{\pm 5.1}$ | 312.8 |
| - expectile in policy update | 93.4 $_{\pm 4.2}$ | 43.4 $_{\pm 26.1}$ | 45.8 $_{\pm 21.8}$ | 11.4 $_{\pm 18.6}$ | 53.8 $_{\pm 7.5}$ | 54.0 $_{\pm 6.9}$ | 33.8 $_{\pm 11.6}$ | 21.4 $_{\pm 13.2}$ | 357.0 |
| + expectile in data transitions | 53.4 $_{\pm 18.8}$ | 4.8 $_{\pm 7.7}$ | 27.2 $_{\pm 37.0}$ | 19.2 $_{\pm 11.9}$ | **69.0** $_{\pm 17.4}$ | **75.4** $_{\pm 6.4}$ | 0.0 $_{\pm 0.0}$ | 0.0 $_{\pm 0.0}$ | 249.2 |

**Ablation on pretraining.**    Table 19 shows the effect of BC and FQE pretraining for LEQ and MOBILE* in AntMaze tasks. Without pretraining, the performance of LEQ decreases (461.8 $\rightarrow$ 322.2), especially in medium mazes (96.4 $\rightarrow$ 4.6) and ultra mazes (81.6 $\rightarrow$ 12.4) due to early instability of training, matching the observation in CBOP (Jeong et al., 2023). However, for MOBILE*, pretraining worsens the performance (285.3 $\rightarrow$ 232.2).

Table 19: **Ablation results for pretraining on AntMaze.** Results are averaged over 5 random seeds.

| Method conservatism | pretrain | umaze | | medium | | large | | ultra | | Total |
|---|---|---|---|---|---|---|---|---|---|---|
| | | umaze | diverse | play | diverse | play | diverse | play | diverse | |
| LEQ | O | **94.4** $_{\pm 6.3}$ | 71.0 $_{\pm 12.3}$ | 50.2 $_{\pm 39.9}$ | **46.2** $_{\pm 23.2}$ | 58.6 $_{\pm 9.1}$ | 60.2 $_{\pm 18.3}$ | 25.8 $_{\pm 18.2}$ | **55.8** $_{\pm 18.3}$ | **461.8** |
| LEQ | X | **94.0** $_{\pm 1.9}$ | 65.6 $_{\pm 6.8}$ | 0.8 $_{\pm 1.6}$ | 3.8 $_{\pm 7.6}$ | 60.4 $_{\pm 13.4}$ | 59.8 $_{\pm 12.5}$ | 11.6 $_{\pm 12.4}$ | 26.2 $_{\pm 22.4}$ | 322.2 |
| MOBILE* | O | 73.4 $_{\pm 12.6}$ | 46.0 $_{\pm 9.4}$ | 24.6 $_{\pm 23.2}$ | 11.6 $_{\pm 9.9}$ | 31.0 $_{\pm 8.4}$ | 33.2 $_{\pm 10.6}$ | 12.4 $_{\pm 6.9}$ | 0.0 $_{\pm 0.0}$ | 232.2 |
| MOBILE* | X | 53.8 $_{\pm 26.8}$ | 22.5 $_{\pm 22.2}$ | 54.0 $_{\pm 5.8}$ | 49.5 $_{\pm 6.2}$ | 28.3 $_{\pm 6.0}$ | 28.0 $_{\pm 11.4}$ | **25.5** $_{\pm 6.9}$ | 23.8 $_{\pm 15.8}$ | **285.3** |

**Ablation study on dataset expansion.**    Table 20 shows the ablation results on the dataset expansion in D4RL MuJoCo tasks. The results show that the dataset expansion generally improves the performance, especially in Hopper environments.

Table 20: **D4RL MuJoCo ablation results for dataset expansion.** Results are averaged over 5 random seeds. The dataset expansion generally improves the performance of LEQ.

| Dataset | LEQ (ours) | LEQ w/o Dataset Expansion |
|---|---|---|
| hopper-r | **32.4** $_{\pm 0.3}$ | 17.6 $_{\pm 8.6}$ |
| hopper-m | **103.4** $_{\pm 0.3}$ | 52.7 $_{\pm 45.3}$ |
| hopper-mr | **103.9** $_{\pm 1.3}$ | 103.7 $_{\pm 1.3}$ |
| hopper-me | **109.4** $_{\pm 1.8}$ | 79.7 $_{\pm 42.4}$ |
| walker2d-r | **21.5** $_{\pm 0.1}$ | 20.5 $_{\pm 2.2}$ |
| walker2d-m | 74.9 $_{\pm 26.9}$ | **87.2** $_{\pm 4.3}$ |
| walker2d-mr | **98.7** $_{\pm 6.0}$ | 78.7 $_{\pm 35.5}$ |
| walker2d-me | 108.2 $_{\pm 1.3}$ | **110.4** $_{\pm 0.8}$ |
| halfcheetah-r | **30.8** $_{\pm 3.3}$ | 27.7 $_{\pm 2.2}$ |
| halfcheetah-m | **71.7** $_{\pm 4.4}$ | 71.6 $_{\pm 3.8}$ |
| halfcheetah-mr | **65.5** $_{\pm 1.1}$ | 54.4 $_{\pm 26.3}$ |
| halfcheetah-me | **102.8** $_{\pm 0.4}$ | 83.9 $_{\pm 28.0}$ |
| **Total** | **923.2** | 788.2 |

**Ablation on using deterministic policy.** We parameterize the stochastic policy $\pi(\cdot|s)$ as $\pi(a|s) = \tanh(N(\mu(s), \sigma^2(s)))$ as Haarnoja et al. (2018) and run LEQ with this configuration. However, we found that the policy quickly becomes deterministic, because LEQ inadvertently penalizes the stochasticity of the policy while penalizing the uncertainty of the model rollouts. Specifically, when we use a stochastic policy, the stochasticity of the policy contributes to increasing the variance of $\lambda$-returns, which are therefore heavily penalized by LEQ.

To compensate for this effect, we use an entropy bonus coefficient $\alpha = 0.0003$. As demonstrated in Table 21, stochastic policy shows slightly worse performance compared to deterministic policy $(461.8 \rightarrow 380.4)$. However, we believe that LEQ can be extended to stochastic policies with further hyperparameter tuning on the stochasticity of the policy.

Table 21: **Ablation results on using deterministic policy.** Results are averaged over 5 random seeds.

| Policy | umaze | | medium | | large | | ultra | | Total |
|---|---|---|---|---|---|---|---|---|---|
| | umaze | diverse | play | diverse | play | diverse | play | diverse | |
| Deterministic | $94.4_{\pm6.3}$ | $71.0_{\pm12.3}$ | $50.2_{\pm39.9}$ | $46.2_{\pm23.2}$ | $58.6_{\pm9.1}$ | $60.2_{\pm18.3}$ | $25.8_{\pm18.2}$ | $55.8_{\pm18.3}$ | 461.8 |
| Stochastic | $91.0_{\pm7.9}$ | $70.4_{\pm6.7}$ | $35.8_{\pm29.7}$ | $0.0_{\pm0.0}$ | $41.8_{\pm16.5}$ | $50.2_{\pm7.8}$ | $43.4_{\pm25.0}$ | $47.8_{\pm23.5}$ | 380.4 |

**Ablation on using learned terminal function.** We conduct an ablation study to evaluate the impact of using learned terminal functions instead of the ground-truth terminal function. For the terminal prediction network, we use a 3-layer MLP with a hidden size of 256, consistent with the architecture of the policy and critic networks. As shown in Table 21, replacing the true terminal function with a learned terminal function leads to a significant drop in performance.

Table 22: **Ablation results on using learned terminal function.** Results are averaged over 5 random seeds.

| Terminal | umaze | | medium | | large | | ultra | | Total |
|---|---|---|---|---|---|---|---|---|---|
| | umaze | diverse | play | diverse | play | diverse | play | diverse | |
| Groundtruth | $94.4_{\pm6.3}$ | $71.0_{\pm12.3}$ | $50.2_{\pm39.9}$ | $46.2_{\pm23.2}$ | $58.6_{\pm9.1}$ | $60.2_{\pm18.3}$ | $25.8_{\pm18.2}$ | $55.8_{\pm18.3}$ | 461.8 |
| Learned | $66.8_{\pm5.2}$ | $44.0_{\pm22.3}$ | $54.0_{\pm31.4}$ | $2.0_{\pm4.0}$ | $29.4_{\pm8.1}$ | $0.0_{\pm0.0}$ | $15.6_{\pm5.9}$ | $20.8_{\pm5.2}$ | 232.6 |

**Ablation study on horizon length in V-D4RL.** Table 23 shows that when we use the default hyperparameter of DreamerV3, $H = 15$, the performance drops in V-D4RL. The result suggests that we need to use a shorter imagination horizon for offline model-based RL.

Table 23: **V-D4RL ablation results for horizon length.** Results are averaged over 3 random seeds. $H = 5$ generally improves the performance of LEQ.

| Dataset | H=5 | H=15 |
|---|---|---|
| walker_walk-random | $23.2_{\pm1.1}$ | $15.4_{\pm1.5}$ |
| walker_walk-medium | $50.0_{\pm3.6}$ | $40.0_{\pm3.6}$ |
| walker_walk-medium_replay | $44.0_{\pm28.5}$ | $45.4_{\pm7.5}$ |
| walker_walk-medium_expert | $90.3_{\pm1.8}$ | $79.9_{\pm1.4}$ |
| cheetah_run-random | $15.3_{\pm1.7}$ | $15.5_{\pm3.0}$ |
| cheetah_run-medium | $40.1_{\pm14.6}$ | $32.2_{\pm4.3}$ |
| cheetah_run-medium_replay | $39.9_{\pm2.0}$ | $36.4_{\pm2.1}$ |
| cheetah_run-medium_expert | $27.7_{\pm12.6}$ | $29.3_{\pm4.1}$ |
| Total | **330.5** | 294.1 |

## F  VIDEO PREDICTIONS ON V-D4RL

Figure 12 presents the imagined trajectory generated by the action sequence during evaluation in the V-D4RL datasets: `walker_walk-medium_replay`, `walker_walk-medium_expert`, `cheetah_run-medium_replay`, `cheetah_run-medium_expert`, from the top to the bottom. Overall, world models trained in `medium_replay` datasets show better prediction compared to `medium_expert` dataset, likely due to their broader state distribution of the dataset. Nevertheless, LEQ achieves high performance on the `walker_walk-medium_expert` dataset, despite inaccurate predictions, highlighting robustness of LEQ in handling imperfect world models.

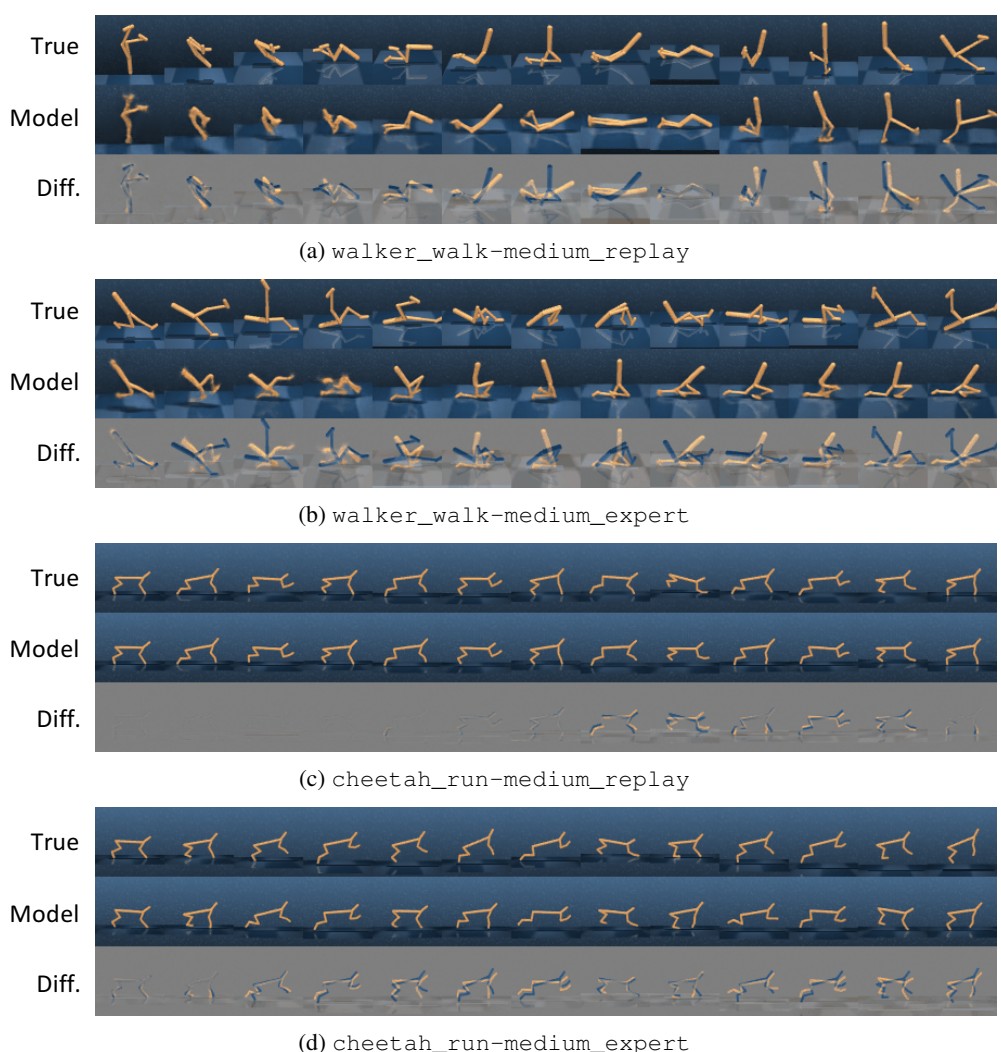

(a) `walker_walk-medium_replay`

(b) `walker_walk-medium_expert`

(c) `cheetah_run-medium_replay`

(d) `cheetah_run-medium_expert`

Figure 12: **Video prediction results in V-D4RL.** The world model receives initial 5 frames and simulates 64 additional frames based on the action sequence during evaluation. For each trajectory, each row displays the true image, model prediction, and difference between these two, respectively.

