# OpenReview forum: "Model-based Offline Reinforcement Learning with Lower Expectile Q-Learning"
_ICLR.cc/2025/Conference — ICLR 2025 Poster_

### Official Review · Reviewer_LGd6 · 2024-10-18

**Soundness:** 3
**Presentation:** 3
**Contribution:** 2
**Rating:** 6
**Confidence:** 5

**Summary:**

This paper introduces a novel model-based offline reinforcement algorithm called Lower Expectile Q-Learning (LEQ). LEQ is similar to prior model-based offline RL algorithms with three main additions. 1) LEQ uses expectile regression for both critic and policy training, to obtain conservative Q-value estimates and actions respectively. 2) LEQ uses $\lambda$-returns for both critic and policy training. 3) The critic is trained with both model-generated and the provided offline data, while the actor is trained with only model-generated data. There are a number of other smaller additions, such as the use of EMA regularization when training the critic and some hyperparameter changes to prior work.

In summary, the approach is clearly explained, comprehensively evaluated and the components are individually studied. Therefore while none of the contributions of the paper are particularly novel or likely to have a major impact on the direction of the field, I believe the paper should be accepted due to the thoroughness of the work. LEQ will likely provide a strong general-purpose baseline for model-based offline RL in future work.

**Strengths:**

- **Clarity**
  - The method is clearly motivated and explained.
  - The results are clearly analyzed and summarized.
  - The paper is generally well-written and structured.
- **Quality and Comprehensiveness**
  - LEQ is comprehensively evaluated across a wide-range of benchmarks. The results are realistically strong all-around, and the authors do a good job of analyzing the strengths of the method. In particular, LEQ appears to perform better than the baselines (impressively both model-based and model-free) at long horizon tasks, such as the larger AntMaze environments.
  - The authors appear to do a good job of ensuring fair comparisons with a wide range of baselines.
  - There are extensive ablation studies to justify the method's contributions and design decisions.
- **Significance**
  - The quality of the work suggests that LEQ will provide a strong and computationally fast baseline for future work on offline RL.

**Weaknesses:**

- **Significance**
  - None of the components of the model are particularly novel, and as a result, I wouldn't expect LEQ to be able to solve any problem that is not already solvable by existing offline RL algorithms. While the all-around improvement is of benefit to the community, this reduces the significance of the work.
  - Other insights such as the benefit of using $\lambda$-returns are generally well-established, and the improvements in hyperparameters are likely to be environment specific and therefore not very generalizable.
  - While standard practice in offline RL, the use of differing hyperparameters per environment (specifically $\tau$, as provided in Table 10), also limits the significance and generality of the work. However, the results for all expectiles provided in the Appendix is helpful and appreciated.
- **Originality**
  - In addition to the novelty of the components, the use of expectile regression as used in IQL for model-free offline RL appears to have already been applied ot model-based offline RL in IQL-TD-MPC [1]. This method appears as a baseline for the AntMaze results, but does not appear in the other results (Tables 2,3,4) or in the Related Work. Given the apparent similarity in the approach, this method should be explained and distinguished in the paper's motivation/related work and ideally used as a baseline for all results. The citation for this method is also wrong in Section 5.3.

**References:**

[1] *IQL-TD-MPC: Implicit Q-Learning for Hierarchical Model Predictive Control*, Chitnis et al. 2024

**Questions:**

- On line 256, the authors state "because of the expectile term ..., we cannot directly compute the gradient... thus we optimize the unnormalized version of the expectile". Please could the authors expand on this line? What is meant by the 'unnormalized' version of the expectile? Also, on line 258 the authors say they approximate the expectation with the learned Q-estimator in order to derive the surrogate loss. But then on line 266 they say the surrogate loss provides a better approximation than the Q-estimator, which naively seems contradictory (without going through Appendix B). Could the authors clarify this please?
- In Equation 7, can you confirm that $\beta$ and $\beta_{EMA}$ are independent hyperparameters? Assuming so, it would be less confusing to use a different symbol for $\beta_{EMA}$.
- Could you explain the main difference between LEQ and IQL-TD-MPC and include this in the paper?
- In Section 5, LEQ is compared with the baselines in terms of performance. It would also be helpful to have some comparison in terms of computational cost/run-time, as this is also an important aspect of these algorithms (and LEQ appears quite strong in this regard from the appendix). Could this be included?
- In Section 5.3, the authors discuss failed trajectories that attempt to pass through walls, and that 'this could be addressed by... improving the number of ensembles for the world models.' This is an interesting failure case, but I found this confusing as my understanding is that LEQ uses a single world model - is this the case? Do the authors have any hypothesis or evidence that using an ensemble would address this issue?
- Similarly in Section 5.4, the authors discuss variance in performance during training due to optimistic prediction. Does a reduced value of $\tau$ in the expectile regression (more conservative estimates) address this issue?
- Section 6.1 on Limitations seems out of place at the end of the work. This seems more appropriate before the conclusion (ideally included as an additional ablation in 5.6 with a minimal termination predictor on one or two environments). Could this paragraph be moved before the conclusion?

---

> ### Author Response · Authors · 2024-11-17
>
> Thank you for your constructive feedback. We address your questions and concerns in detail below and our new manuscript reflects your suggestions. Please refer to the updated PDF for new results and revisions.
>
> &nbsp;
>
> **[W1, W2-1] None of the components of the model are particularly novel, and as a result, I wouldn't expect LEQ to be able to solve any problem that is not already solvable by existing offline RL algorithms. Other insights such as the benefit of using λ-returns are generally well-established.**
>
>
> First, we propose a novel methodology to get conservative return estimates via expectile regression. Please note that **application of expectile regression in LEQ fundamentally differs from IQL**; LEQ utilizes **lower** expectile regression to get conservative return estimates from potentially inaccurate model rollouts, while IQL employs **upper** expectile regression to approximate the max operation in $V(s) = \max_{a} Q(s, a)$.
> Moreover, we introduce a **novel surrogate loss for policy optimization**, designed to maximize the expectile of sampled returns, previously unexplored in the field. Although prior works have demonstrated the effectiveness of $\lambda$-returns, our extension of policy gradients for $\lambda$-returns to expectile is novel and crucial for offline MBRL. We believe this innovation will be beneficial for future offline MBRL research.
> Finally, LEQ is able to solve AntMaze tasks, which is **nearly unsolvable for previous model-based methods**. Specifically, most of the model-based offline RL shows zero score in AntMaze tasks; IQL-TD-MPC is the only model-based algorithm showing non-zero scores in AntMaze tasks, but still much worse than LEQ in AntMaze as well as HalfCheetah.
>
> &nbsp;
>
> **[W2-2, W3] The improvements in hyperparameters are likely to be environment specific and therefore not very generalizable. While standard practice in offline RL, the use of differing hyperparameters per environment (specifically τ, as provided in Table 10), also limits the significance and generality of the work.**
>
> We first would like to emphasize that **LEQ with $\tau=0.1$ consistently shows the strong performance** across all benchmarks, as shown in Table 15-17 in Appendix C.
>
> Moreover, **LEQ shares the same hyperparameters across all state-based environments**, except a single hyperparameter $\tau$ for penalization. In contrast, prior works such as MOBILE and MOPO not only tune the penalization coefficient but also search for the optimal rollout length, with different hyperparameter ranges for different domains.
>
> &nbsp;
>
> **[W4-1, Q3] Could you explain the main difference between LEQ and IQL-TD-MPC and include this in the paper?**
>
> While both LEQ and IQL-TD-MPC are model-based offline RL methods that utilize expectile regression, these two approaches are **fundamentally different**. Specifically, IQL-TD-MPC does not penalize inaccurate model rollouts as LEQ. Instead, it mitigates model inaccuracies by removing random actions from MPPI and constraining the policy search to actions near the initial AWR-learned policy. This effectively limits the policy to predict actions similar to those present in the dataset, similar to IQL.
>
> Accordingly, IQL-TD-MPC employs **upper** expectile regression to **approximate the max operation** in $V(s) = \max_{a} Q(s, a)$, for calculating the advantage for AWR, while LEQ utilizes **lower** expectile regression to get **conservative return estimates** from potentially inaccurate model rollouts.
>
> We have clarified the differences between LEQ and IQL-TD-MPC in L121-125 of Section 2 and included their comparisons in Table 3 using HalfCheetah tasks, where official scores for IQL-TD-MPC are available. Notably, LEQ significantly outperforms IQL-TD-MPC in HalfCheetah tasks (240.0 vs 151.4).
>
> &nbsp;
>
> **[W4-2] The citation for IQL-TD-MPC is also wrong in Section 5.3**
>
>
> Thank you for pointing out our mistake. We corrected the citation for IQL-TD-MPC in L363 of Section 5.3 in the revised paper.

---

> ### Author Response · Authors · 2024-11-17
>
> **[Q1-1] On line 256, the authors state "because of the expectile term ..., we cannot directly compute the gradient... thus we optimize the unnormalized version of the expectile". Please could the authors expand on this line? What is meant by the 'unnormalized' version of the expectile**
>
>
> Thank you for pointing this out. From the definition, $\mathbb{E}^{\tau}[X]$, expectile of a distribution $X$, is a minimizer of $L(y) = \mathbb{E}\_{x \sim X}[|\tau - 1(y>x)| (y-x)^2]$. Since the gradient of $L(\mathbb{E}^{\tau}[X])$ is zero (because $\mathbb{E}^{\tau}[X]$ is the minimum), we have the formula $L'(\mathbb{E}^{\tau}[X]) = \mathbb{E}\_{x \sim X}[|\tau - 1(\mathbb{E}^{\tau}[X]>x)| \cdot 2 \cdot (\mathbb{E}^{\tau}[X]-x)|] = 0$. By rearranging the equation, we can have $\mathbb{E}^{\tau}[X] = \frac{\mathbb{E}\_{x \sim X}[|\tau - 1(\mathbb{E}^{\tau}[X]>x)| \cdot x]}{\mathbb{E}\_{x \sim X}[|\tau - 1(\mathbb{E}^{\tau}[X]>x)|]}$. We remove the normalizer (denominator) and refer $\mathbb{E}\_{x \sim X}[|\tau - 1(\mathbb{E}^{\tau}[X]>x)| \cdot x]$ as “unnormalized version of the expectile”.
>
> We added this explanation on the derivation of surrogate loss and clarified the meaning of the terms in L257-258 of Section 4.3.
>
> &nbsp;
>
> **[Q1-2] Also, on line 258 the authors say they approximate the expectation with the learned Q-estimator in order to derive the surrogate loss. But then on line 266 they say the surrogate loss provides a better approximation than the Q-estimator, which naively seems contradictory (without going through Appendix B). Could the authors clarify this please?**
>
> To clarify, L266-267 means that minimizing our surrogate loss (consists of $Q\_\phi(\mathbf{s}\_t, \mathbf{a}\_t)$ and $Q^\lambda\_t(\mathcal{T})$) leads to a better policy than directly minimizing $-Q\_\phi(\mathbf{s}\_t, \mathbf{a}\_t)$. This is not contradictory.
>
> Specifically speaking, the learned Q-estimator provides a high-bias, low-variance estimate, while the surrogate loss derived from $\lambda$-returns serves as a low-bias, high-variance estimator. When learned Q-values are inaccurate, using $\lambda$-returns offers a better approximation of the true Q-values by relying on a multi-step return computed (although it also uses the inaccurate learned Q-values instead of true Q-values), reducing bias at the cost of increased variance. We note that if $\tau = 0.5$, the policy optimization is equivalent to optimizing the $\lambda$-returns instead of Q-values.
>
>
>
> **[Q2] In Equation 7, can you confirm that $\beta$ and $\beta_{\text{EMA}}$ are independent hyperparameters? Assuming so, it would be less confusing to use a different symbol for $\beta_{\text{EMA}}$.**
>
> Yes, those are independent hyperparameters. To make it clearer, we changed the notation $\beta_{\text{EMA}}$ to $\omega_{\text{EMA}}$. Thank you for your suggestion!
>
> &nbsp;
>
> **[Q3] ​​Could you explain the main difference between LEQ and IQL-TD-MPC and include this in the paper?**
>
> Please refer to **[W4-1, Q3]**.
>
> &nbsp;
>
>
> **[Q4] In Section 5, LEQ is compared with the baselines in terms of performance. It would also be helpful to have some comparison in terms of computational cost/run-time, as this is also an important aspect of these algorithms (and LEQ appears quite strong in this regard from the appendix). Could this be included?**
>
>
> Thank you for the suggestion. Unfortunately, we did not have enough space for the main paper to include a comparison table for the computational cost in Section 5. Instead, we compared the run-time with MOBILE and CBOP in L310-311 of Section 5.2, and referred to Table 8 of Appendix A. Notably, LEQ is $6$x faster than recent model-based offline RL algorithms like CBOP and MOBILE.
>
> &nbsp;
>
> **[Q5-1] I found this confusing as my understanding is that LEQ uses a single world model - is this the case?**
>
> Following MOBILE, LEQ uses an ensemble of 5 world models by selecting 5 out of 7 models with the best validation score. Please refer to L732-735 of Appendix A for further details.
>
> &nbsp;
>
> **[Q5-2] In Section 5.3, the authors discuss failed trajectories that attempt to pass through walls, and that 'this could be addressed by ... improving the number of ensembles for the world models.' … Do the authors have any hypothesis or evidence that using an ensemble would address this issue?**
>
> Our hypothesis for the AntMaze-medium maze is that if 1) at least one of the world models can predict that the wall is impenetrable and 2) we apply sufficiently low $\tau$, LEQ will correctly evaluate the returns (since it will value the most conservative model prediction). Thus, we believe that better world models (which can predict that the wall is blocked) or increasing the number of world models (so that at least one of the world models can predict that the wall is  impenetrable) can address the issue.

---

> ### Author Response · Authors · 2024-11-17
>
> **[Q6] Similarly in Section 5.4, Does a reduced value of $\tau$ in the expectile regression (more conservative estimates) address this issue?**
>
> As shown in Table 16 of Appendix C, reducing the value of $\tau$ helps decrease the standard deviation in Hopper tasks. However, in Walker2D tasks, fluctuations persist, resulting in a large standard deviation even with a lower $\tau$. This occurs because all the world models consistently predict overly optimistic transitions, similar to the failure cases observed in AntMaze.
>
> &nbsp;
>
> **[Q7] Placement of Section 6.1**
>
> Thank you for the suggestion! Replacing the true terminal function with a learned terminal function leads to a significant performance drop ($461.8 \rightarrow 232.6$), especially in diverse datasets ($233.2 \rightarrow 66.8$), where termination signals are scarce because the dataset is collected by navigating to randomly selected goals.
>
> We added the experimental results and their setup in Table 22 of Appendix D. We moved the limitation to Section 6 (between the experiments and conclusion sections) and explained these results in L520-522 of Section 6.
>
> &nbsp;
>
> **References**
>
> [1] Hafner et al. "Mastering diverse domains through world models.", Arxiv 2023.

---

> > ### Comment · Reviewer_LGd6 · 2024-11-22
> > **Thank you to the authors for their response and improvements to the paper**
> >
> > Thank you to the authors for their thorough response and highlighting the changes to the paper. Regarding the responses to my main questions:
> >
> > **Q1.** This is now clear and I appreciate the inclusion in the paper, which helps to clarify the method for lower expectile policy learning.
> >
> > **Q3.** Thank you for clarifying the difference with IQL-TD-MPC and IQL and the inclusion of this in the paper. I now better understand that this enables more flexibility in the policy learning and should be less reliant on behavior cloning, likely leading to the improved results. I also appreciate the inclusion of the IQL-TD-MPC results on D4RL, but given the lack of coverage and explanation of the differences, unless the authors are able to run this baseline on the other D4RL environments, I can now appreciate that this baseline is not worth including in Table 3 and could be removed.
> >
> > **Q4.** I believe this is an important addition, and believe this significantly strengthens the contribution of LEQ, so appreciate the inclusion.
> >
> > **Q5.** Thank you for this clarification. While I now see that this is assumed to follow from MOBILE as an implementation detail, I do feel a short mention of this would be helpful in the main text as well as the addition in the appendix. Additionally, regarding the benefits of pre-training the policy and critic with behavior cloning and FQE explained in this appendix, a reference to *Efficient Offline Reinforcement Learning: The Critic is Critical*, Jelley et al. (2024), which demonstrates efficiency and stability improvements in off-policy RL from pre-training could be helpful justification for this stage.
> >
> > **Q6.** Thank you for the explanation. This is clear, and I now agree that the robustness to hyperparameters (with $\tau=0.1$ as shown in Appendix C) is a strength of the approach.
> >
> > **Q7.** This rearrangement and addition of limitations helps to improve the flow of the end of the paper, so is appreciated.
> >
> > Overall, I believe this paper provides a meaningful contribution the offline reinforcement learning community, as well as a robust model-based baseline, and so deserves acceptance. Since a score of 7 is not available, I will maintain my score of 6 but have increased my confidence to 5 that it should be accepted.

---

> > > ### Author Response · Authors · 2024-11-23
> > >
> > > We appreciate your prompt reply and additional suggestions.
> > >
> > > **Q3.** We removed the result of IQL-TD-MPC in Table 3.
> > >
> > > **Q5.** Thank you for your suggestion! We further updated our paper, mentioning MOBILE for the design choices in L309-310 in Section 5 and adding the reference to (Jelley et al., 2024) in L750 in Appendix A.
> > >
> > > Thank you again for your valuable feedback and for increasing the confidence score!

---

### Official Review · Reviewer_2jfa · 2024-10-19

**Soundness:** 3
**Presentation:** 3
**Contribution:** 3
**Rating:** 6
**Confidence:** 5

**Summary:**

To to address the uncertainty of model data $Q$ value estimation, the authors  introduce low expectile regression into the algorithm, and propose a novel model-based offline RL method. This method can achieve conservative estimation for model data and demonstrates superior performance across multiple datasets.

**Strengths:**

1) The article is well-written overall, with clear logic and a detailed introduction on how to incorporate low expectile regression into RL algorithm. The experiments are thorough, validating the algorithm's superiority across multiple datasets.

2) The methodology presented in the article is very intriguing. Unlike traditional value penalization methods, this paper innovatively employs low expectile regression to alleviate the issue of inaccurate model data estimation.

**Weaknesses:**

1) Theoretical: The article does not provide the theoretical analysis of how the use of low expectile regression impacts the final Q-values of the model data. Additionally, it does not offer theoretical guarantees for policy improvement , as seen in algorithms like COMBO and MOBILE.

2) Algorithm: The method presented in this paper directly scales the loss values of the overestimated model data by a factor of $\tau$, without considering the differences among the model data. For example, the scaling coefficient $\tau$ should vary depending on the degree of error in the model data.

**Questions:**

1) Does the addition of the third loss term in Eq. (7) affect final performance? The authors could provide simple experiments for analysis.

2) Why does the policy update use only the generated model data and not the precise offline data? Most model-based offline RL algorithms currently use accurate offline data to update policy.

3) Is there an error about the definition of $L_2^{\tau}$ in Eq. (1)? According to my understanding, in Eq. (4), $L_2^{\tau}(u)=\|\tau-\mathbb{I}(u>0)\|u^2$.

---

> ### Author Response · Authors · 2024-11-17
>
> Thank you for your constructive feedback. We address your questions and concerns in detail below and our new manuscript reflects your suggestions. Please refer to the updated PDF for new results and revisions.
>
> &nbsp;
>
> **[W1] The article does not provide the theoretical analysis of how the use of low expectile regression impacts the final Q-values of the model data. Additionally, it does not offer theoretical guarantees for policy improvement , as seen in algorithms like COMBO and MOBILE.**
>
> Our extensive ablation studies in Table 5 demonstrate the impact of lower-expectile regression and the surrogate loss on policy improvement. Specifically, replacing lower-expectile Q-learning with MOBILE penalization leads to a substantial performance drop in AntMaze tasks ($461.8 \rightarrow 338.9$). Similarly, using $Q(s,a)$ ($461.8 \rightarrow 373.5$) or AWR ($461.8 \rightarrow 114.2$) in place of our surrogate loss for policy optimization results in notable performance degradation. These empirical findings support our claim that lower-expectile Q-learning effectively mitigates inaccuracies in model rollouts and that our surrogate loss better improves the policy over prior approaches.
>
> We agree that theoretical analysis can definitely enhance our paper. However, we believe our comprehensive empirical results provide valuable insights into offline RL and can serve as a foundation for future research. We believe that these empirical findings can sufficiently support our claims, despite the lack of theoretical support.
>
> &nbsp;
>
> **[W2]  The method presented in this paper directly scales the loss values of the overestimated model data by a factor of $\tau$, without considering the differences among the model data. For example, the scaling coefficient $\tau$ should vary depending on the degree of error in the model data.**
>
> To clarify, our method already penalizes states with high model errors more strongly than states with low model errors. This is because high uncertainty in model predictions results in increased variance in the target returns, leading to high variance in the target returns. For example, if the target returns follow the normal distribution, lower-expectile is equivalent to lower-confidence bound (LCB), which penalizes the mean estimate with its standard deviation.
>
> We agree that developing a mechanism that further adapts the conservatism $\tau$ based on state-specific information is an exciting direction for future research. We do not have a good idea about how to implement it yet, and we note that similar limitations also exist in prior approaches, such as MOBILE, which applies a single LCB coefficient uniformly across all states, regardless of their individual characteristics.
>
> &nbsp;
>
> **[Q1] Does EMA regularization affect final performance?**
>
> Table 18 in Appendix D shows that replacing EMA regularization with a target network leads to a decrease in performance ($461.8 \rightarrow 379.1$), with an increase in Bellman error. Although we believe that further hyperparameter tuning can improve the performance of the one with a target network,we opted for EMA regularization since it works well without tuning.
>
> &nbsp;
>
> **[Q2] Why does the policy update use only the generated model data and not the precise offline data? Most model-based offline RL algorithms currently use accurate offline data to update policy.**
>
>
> While training a policy using both offline data and model-generated data (similar to the critic training in LEQ) could improve the policy, we opted for the simpler Dreamer-style policy update, which we found to be sufficient in practice.
>
> &nbsp;
>
> **[Q3] Error about the definition of $L_2^{\tau}$ in Eq. (1)**
>
> Thank you for pointing this out. We fixed the definition of $L_2^{\tau}$ in Eq. (1) and the explanation in L153-157 of Section 3 accordingly.

---

> ### Comment · Reviewer_2jfa · 2024-11-24
>
> Thank you very much for the author's reply. The author conducted a lot of experiments to verify the effectiveness of the algorithm. I think this article is acceptable. However, due to the defects in the theoretical analysis, I still maintain my score.

---

### Official Review · Reviewer_xSxW · 2024-10-29

**Soundness:** 2
**Presentation:** 3
**Contribution:** 2
**Rating:** 6
**Confidence:** 4

**Summary:**

This paper introduces Lower Expectile Q-learning (LEQ), a novel model-based offline reinforcement learning method that utilizes lower expectile of Q values to avoid overestimation.  The success of LEQ is attributed to lower expectile regression, λ-returns, the lower expectile target for the policy training. LEQ significantly outperforms previous MBRL methods on long-horizon tasks and matches state-of-the-art model-free methods in various environments.

**Strengths:**

- The paper is well-written and easy to follow. The description to the thought of design and the algorithm is clear.
- The paper provides both experimental and theoretical analysis, which enhances the soundness of the lower expectile policy learning.
- The experiments are conducted on various types of tasks, including dense- and sparse-reward environments, along with state and image input formats, showing the effectiveness of LEQ.

**Weaknesses:**

See questions.

**Questions:**

- I do not find any reference of Figure 2 in the paper. Could you please insert the figure into the proper place to better explain your method?
- How do you use the ensemble of world models for sampling? For example, do you randomly select one model for each step or for each rollout?
- For the experiments, though you include TT and TAC as baselines, I do not find any analysis of the related results. Could you please explain why to include TT and TAC, and provide what we can learn from comparing LEQ against them?
- In the proof of Theorem 1, it seems that the last inequality holds if $\mathbb{E}[W^\tau(Y>X)] \ge 1-\tau$. However, when $0<\tau \le 0.5$, we have $\tau \le W^\tau(\cdot) = |\tau - 1(\cdot)| \le 1-\tau$, so $\tau \le \mathbb{E}[W^\tau(\cdot)] \le 1-\tau$. As a result, the last inequality might be in the wrong direction.
- What is the difference between MBPO and Dyna training schemes, as you classify in Table 8? Previous works usually regard MBPO as a Dyna-style algorithm. For example, though you classify MOBILE into the MBPO style, MOBILE itself claims that "MOBILE is under the Dyna-style framework" and cite MBPO as a "Dyna-style model-based RL" method in Section 5 of its original paper[1].

[1] Model-Bellman Inconsistency for Model-based Offline Reinforcement Learning. Yihao Sun, Jiaji Zhang, Chengxing Jia, Haoxin Lin, Junyin Ye, Yang Yu. Proceedings of the 40th International Conference on Machine Learning, PMLR 202:33177-33194, 2023.

---

> ### Author Response · Authors · 2024-11-17
>
> Thank you for your constructive feedback. We address your questions and concerns in detail below and our new manuscript reflects your suggestions. Please refer to the updated PDF for new results and revisions.
>
> &nbsp;
>
> **[Q1] I do not find any reference of Figure 2 in the paper. Could you please insert the figure into the proper place to better explain your method?**
>
> Thank you for your suggestion. We added the references to Figure 2 in L120 and L213 in the revised version.
>
> &nbsp;
>
> **[Q2] How do you use the ensemble of world models for sampling? For example, do you randomly select one model for each step or for each rollout?**
>
> We select one model for each step, following the design choice of MOBILE [1]. We found that randomly choosing a model every step can make imaginary rollouts more robust to model biases, leading to better performance.
>
> We clarified this in L739-L741 of Appendix A in the revised paper.
>
> &nbsp;
>
> **[Q3] Could you please explain why to include TT and TAP, and provide what we can learn from comparing LEQ against them?**
>
> Sequence modeling approaches (e.g. TT, TAP) serve as strong baselines on long-horizon tasks, such as AntMaze. We compare LEQ with these approaches and show that LEQ outperforms them in the most challenging task, AntMaze-Ultra. Moreover, while sequence modeling methods struggle on dense-reward tasks, LEQ maintains near-SOTA performance, demonstrating its broad applicability.
>
> We clarified this in L367-368 of Section 5.3 and L413-414 of Section 5.4 in the revised paper.
>
> &nbsp;
>
> **[Q4] Error in the proof (last inequality)**
>
> We greatly appreciate that you pointed out our mistake in the proof in Appendix B. We could revise the proof that still ensures our original claim--$|Y - \hat{Y}\_\textrm{new}| \leq |Y-\hat{Y}|$, which means **the proposed estimator in our surrogate loss is better than the learned estimator (Q).** The revised theorem now have a looser bound $|Y - Y\_\textrm{new}| \leq (\frac{1-2\tau}{1-\tau}) * |Y-\hat{Y}|$ (previously $|Y - Y_\textrm{new}| \leq (\frac{1-2\tau}{1-\tau}) * p(Y \leq X \leq \hat{Y}) * |Y-\hat{Y}|$); but, this is still sufficient to support our original claim.
>
> &nbsp;
>
> **[Q5] What is the difference between MBPO and Dyna training schemes, as you classify in Table 8? Previous works usually regard MBPO as a Dyna-style algorithm.**
>
> Thank you for pointing out our mistakes again. We agree that describing our method’s training scheme as Dyna is not clear as MBPO is also a Dyna-style algorithm. We originally used Dyna to represent a Dyna-style training scheme but not MBPO. LEQ follows a Dreamer-style training scheme [2, 3], which uses the model for both expanding the dataset and calculating the target values ($\lambda$-returns), while MBPO only uses the model for dataset expansion.
>
> To clearly distinguish LEQ from CBOP and MOBILE, we changed the “training scheme” of LEQ in Table 8 from Dyna to Dreamer.
>
> &nbsp;
>
> **References**
>
> [1] Sun et al. "Model-Bellman inconsistency for model-based offline reinforcement learning.”, ICML 2023.
>
> [2] Hafner et al. "Mastering Atari with Discrete World Models.", ICLR 2021.
>
> [3] Hafner et al. "Mastering diverse domains through world models.", Arxiv 2023.

---

> ### Comment · Reviewer_xSxW · 2024-11-24
>
> I am very sorry for my late reply. Thanks for your answers which have cleared most of my confusion. However, after careful checking, I still have quesiton about the revised proof to Theorem 1. The deduction from Line 954 to Line 956 seems using $E[W^\tau(Y>X)] = \tau + (1-2\tau) p(Y \le X)$. This might still be wrong, as
> $$
> E[W^\tau(Y>X)] = E[|\tau - 1(Y>X)|] = P(Y \le X)\tau + (1-P(Y \le X))(1-\tau) = 1 - \tau - (1-2\tau)P(Y \le X)
> $$
>
> In fact, here may be a counterexample to Theorem 1. Suppose $X$ follows the uniform distribution over $[0,1]$, then $Y = E^\tau[X] = \tau$. Let $\hat Y=0$ be an estimate of $Y$, resulting in $W^\tau(\hat Y > X)=\tau$ for any $X$. Now we can directly compute $\hat Y_{new}$ as
> $$
> \hat Y_{new} = \frac{E[W^\tau(\hat Y > X)X]}{E[W^\tau(\hat Y > X)]} = \frac{\tau E[X]}{\tau} = 0.5~.
> $$
> If $0 < \tau < 0.25$, we have $|\hat Y_{new} - Y| = |0.5 - \tau| > 0.25 > |0 - \tau| = |\hat Y - Y|$, which violates Theorem 1.
>
> I think the motivation and the experiment results of LEQ are quite convincing. So if I misunderstood your proof or you could give a new proof, I will follow up as soon as possible.

---

> > ### Author Response · Authors · 2024-11-26
> >
> > Thank you for your thoughtful response and for providing the counterexample. We apologize that Theorem 1 has an error that our surrogate estimator is not always better than the original estimator, $Q(\mathbf{s}, \mathbf{a})$. Specifically, the inequality $|\hat{Y}_{\text{new}} - Y| \leq |\hat{Y} - Y|$ does not hold everywhere.
> >
> > However, we found that **these exceptions are very rare**.
> > We empirically show this in Figure 7 of Appendix B, the inequality holds always in normal distribution, and 99.9% of the cases in uniform distribution, where remaining 0.1% includes the reviewer xSxW's counterexample when $\hat{Y} = 0$.
> > Theoretically, the inequality breaks only if $p(X \leq \hat{Y}) \leq \frac{1}{2} (p(X \leq Y) - \frac{\tau}{1-2\tau})$.
> > This condition seldom holds because the right-hand side term requires both $\tau$ to be small and $p(X \geq Y)$ to be large, which is contradicting to each other.
> >
> > Although $|\hat{Y}_{\text{new}} - Y| \leq |\hat{Y} - Y|$ holds in most cases and LEQ generally works well, such rare exceptions might have contributed to training instabilities, which we have observed in Section 5.3 and 5.4. We are currently working on finding a better way of optimizing the policy with lower expectile Q-learning.
> >
> > We have updated the theorem and proof with the contents mentioned above, and added a figure illustrating the empirical regions of exceptional cases in Appendix B.
> >
> > We greatly appreciate your thorough review on our proofs. This greatly helps us correct and improve our paper.

---

> > > ### Author Response · Authors · 2024-11-26
> > >
> > > For the future readers, we kindly wish to add a minor clarification regarding the counterexample:
> > >
> > > $Y = E^{\tau}[X] = \frac{-\tau + \sqrt{\tau - \tau^2}}{1-2\tau}$, as it must satisfy the equation: $\mathbb{E}[W^{\tau}(Y > X) (Y - X)] = (1 - \tau) Y^2 - \tau (1 - Y)^2 = 0$.

---

> > > ### Comment · Reviewer_xSxW · 2024-11-26
> > >
> > > Thanks for your reply and I apologize for the mistake in the counterexample above. I believe the new version of Theorem 1, including Lemma 1 and 2, is more solid now. A minor suggestion is that you can add a few lines to separate the case $\tau=0.5$ from the conditions of Lemma 2 and Theorem 1 as $\tau=0.5$ will result in zero division in $\frac{\tau}{1-2\tau}$.
> > >
> > > With the theoretical errors fixed, I think this is generally a good paper and am willing to raise my score to 6.

---

### Official Review · Reviewer_FFdb · 2024-11-03

**Soundness:** 3
**Presentation:** 4
**Contribution:** 3
**Rating:** 6
**Confidence:** 5

**Summary:**

This paper proposes a lower expectile Q-learning (LEQ) approach to address the overestimation issue in model-based offline RL. The LEQ loss assigns higher weight to lower Q-target values and lower weight to higher Q-target values. Through this training mechanism, LEQ achieves conservative value function estimation without the need for uncertainty estimation. Additionally, the authors introduce the $\lambda$-return as the Q-target and the policy update objective. Extensive experiments on offline RL tasks demonstrate that LEQ consistently outperforms previous baselines.

**Strengths:**

1. LEQ is a simple yet highly effective algorithm, easy to implement, and does not introduce significant computational complexity. It exhibits substantial performance advantages across a range of offline RL tasks.
2. Beyond its impressive performance improvements, the authors provide in-depth analysis of LEQ’s instability on several tasks, giving readers a comprehensive understanding of the algorithm and hinting at multiple future research directions.
3. The paper includes extensive performance comparisons and ablation studies, offering thorough insights into the contributions of each component of the algorithm.

**Weaknesses:**

I would greatly appreciate clarification on the following points:

1. Why did the authors choose to update the policy based on deterministic policy gradients? What considerations influenced this choice, and are there experimental differences between using a stochastic versus deterministic policy?
2. In Algorithm 1, the authors pretrain the policy with BC using the offline data, which is not typically done in algorithms like MOPO or MOBILE. How does BC pretraining impact the results, particularly in tasks like AntMaze?

**Questions:**

See weaknesses.

---

> ### Author Response · Authors · 2024-11-17
>
> Thank you for your constructive feedback. We address your questions and concerns in detail below and our new manuscript reflects your suggestions. Please refer to the updated PDF for new results and revisions.
>
> &nbsp;
>
> **[W1] Why did the authors choose to update the policy based on deterministic policy gradients? What considerations influenced this choice, and are there experimental differences between using a stochastic versus deterministic policy?**
>
> Thank you for pointing this out. We used a deterministic policy since lower-expectile regression of LEQ inadvertently penalizes the stochasticity of the policy while penalizing the uncertainty of model rollouts.
>
> Empirically, LEQ with stochastic policies shows worse performance compared to LEQ with deterministic policies in AntMaze ($461.8 \rightarrow 380.4$). Although the performance of LEQ with stochastic policies could be further improved by increasing the stochasticity bonus (e.g. entropy), we suggest using a deterministic policy in LEQ for its simplicity.
>
> We included the corresponding experimental results and their setup in Table 21, Appendix D.
>
> &nbsp;
>
> **[W2] How does BC pretraining impact the results, particularly in tasks like AntMaze?**
>
> This is a good point! Without BC pretraining, the performance of LEQ decreases, $461.8 \rightarrow 322.2$ in total. This is more apparent especially in medium mazes ($96.4 \rightarrow 4.6$) and ultra mazes ($81.6 \rightarrow 12.4$) due to early instability of training, matching the observation in CBOP [1].
>
> In addition, we also conducted additional experiments on MOBILE$^*$ with BC training. However, the result shows that MOBILE$^*$ with BC pretraining degrades the performance ($285.3 \rightarrow 232.2$). We believe that careful initialization of the networks can mitigate this issue.
>
> We added the corresponding experimental results in Table 19, Appendix D.
>
>
> &nbsp;
>
> **Reference**
>
> [1] Jeong et al. "Conservative Bayesian Model-Based Value Expansion for Offline Policy Optimization.", ICLR 2023.

---

> > ### Comment · Reviewer_FFdb · 2024-11-22
> >
> > Thank you very much for your response. After reading your rebuttal and the other reviews, I am more convinced that this paper meets the standards of the ICLR conference. As a result, I have decided to raise my confidence score to 5. Here are a few points for discussion:
> >
> > 1. **"LEQ inadvertently penalizes the stochasticity of the policy."**  Could you elaborate on why LEQ penalizes the stochasticity of the policy?
> >
> > 2. **"Pretraining"**  This is a interesting result. In my view, it is quite challenging to explain why BC pretraining has such a significant impact, especially considering its opposite effects on MOBILE and LEQ. This likely requires a more in-depth analysis, though I understand this may be beyond the scope of this paper.

---

> > > ### Author Response · Authors · 2024-11-23
> > >
> > > Thank you for your reply to our rebuttal. We are glad to hear that you are more convinced with our rebuttal. We further address your additional questions.
> > >
> > > **[Q1] "LEQ inadvertently penalizes the stochasticity of the policy"**
> > >
> > > Our lower-expectile Q-learning (in some sense) selects $\tau$-expectile (relatively low) target Q-value among target Q-values computed from multiple imaginary trajectories with the uncertainties in models. This, in other words, penalizes Q-values with high uncertainties (or inaccuracies) in models.
> > >
> > > If we use a stochastic policy, it will lead to even more diverse imaginary trajectories due to the additional randomness from the policy. Then, LEQ tends to learn even lower Q-values, which is essentially penalizing Q-values with high uncertainties in the policy.
> > >
> > > We clarified this point in L1084-1085 of Appendix D.
> > >
> > > **[Q2] "Pretraining"**
> > >
> > > We also found this controversial result interesting; but, we have not figured out a clear explanation on this result.
> > > We would love to demystify the effect of the BC pretraining as a future work.
> > >
> > > Thank you for your invaluable feedback and suggestions! Hope our response addresses all your concerns. Please let us know if there is any remaining concern that prevents you from increasing the score.

---

### Author Response · Authors · 2024-11-17
**Response to all reviewers**

We thank all reviewers for their constructive feedback! We included new experimental results and revisions in the PDF following the reviewers’ suggestions. Please check out our revised paper. We highlighted the changes in the revised version.

Here, we summarize the major changes:

* Additional experiments on the use of a stochastic policy (Table 21, Appendix D)
* Additional ablation study on BC pretraining of LEQ and MOBILE (Table 19, Appendix D)
* Additional experiments with the learned termination function (Table 22, Appendix D)
* Correction of the proof in Appendix B, the reference of the IQL-TD-MPC paper, and ambiguous descriptions

---

### Meta-Review · Area_Chair_amAR · 2024-12-24

**Metareview:**

The paper introduces Lower Expectile Q-learning (LEQ), a model-based offline RL algorithm that uses lower expectile regression of multi-step returns to mitigate Q-value overestimation from model-generated rollouts. Empirically, LEQ outperforms prior model-based approaches on challenging tasks (notably AntMaze) and delivers competitive or SOTA results on locomotion and visual tasks (e.g., D4RL, NeoRL, V-D4RL).

MBRL methods can use calibrated model uncertainty, which in my opinion is a more principled approach than using expectile regression. However, it is very interesting that a simpler method works better in practice. The paper includes a direct ablation to such a method (MOBILE) and shows that it is not as good as LEQ. This is interesting and makes me think that there is more to be explored in this direction. Perhaps a bigger limitation is that there is lack of theoretical grounding of “how the use of low expectile regression impacts the final Q-values of the model data” (as per reviewer 2jfa). Authors acknowledge this problem and say that a proxy for this is “Our extensive ablation studies in Table 5 demonstrate the impact of lower-expectile regression and the surrogate loss on policy improvement”. The experimental comparisons are thorough and the AntMaze task is quite hard but I would have liked to see more extensive experimental validation on diverse tasks. But there was consensus that this is acceptable and enough to make the core point.

**Additional Comments On Reviewer Discussion:**

LEQ is simple, effective and demonstrates strong empirical results. The baselines and ablations are thorough. Despite some questions around theoretical completeness, the method is sufficiently grounded and empirically successful. Several of the techniques — expectile regression, multi-step returns, and conservative model rollouts — are considered established approaches in offline RL and cannot be considered novel contributions. But this is not a real problem and the overall combined analysis/demonstrations are interesting and deeply meaningful. This is perhaps why all reviewers wanted to accept it but did not strongly advocate for the paper.

---

### Decision · Program_Chairs · 2025-01-22

Accept (Poster)